# Cross-Modal Representational Knowledge Distillation for Enhanced Spike-Informed LFP Modeling

**Eray Erturk**[1]    **Saba Hashemi**[2]    **Maryam M. Shanechi**[1,2,3,4*]
[1]Ming Hsieh Department of Electrical and Computer Engineering
[2]Thomas Lord Department of Computer Science
[3]Alfred E. Mann Department of Biomedical Engineering
[4]Neuroscience Graduate Program
University of Southern California, Los Angeles, CA
{eerturk, saba.hashemi, shanechi}@usc.edu

## Abstract

Local field potentials (LFPs) can be routinely recorded alongside spiking activity in intracortical neural experiments, measure a larger complementary spatiotemporal scale of brain activity for scientific inquiry, and can offer practical advantages over spikes, including greater long-term stability, robustness to electrode degradation, and lower power requirements. Despite these advantages, recent neural modeling frameworks have largely focused on spiking activity since LFP signals pose inherent modeling challenges due to their aggregate, population-level nature, often leading to lower predictive power for downstream task variables such as motor behavior. To address this challenge, we introduce a cross-modal knowledge distillation framework that transfers high-fidelity representational knowledge from pretrained multi-session spike transformer models to LFP transformer models. Specifically, we first train a teacher spike model across multiple recording sessions using a masked autoencoding objective with a session-specific neural tokenization strategy. We then align the latent representations of the student LFP model to those of the teacher spike model. Our results show that the Distilled LFP models consistently outperform single- and multi-session LFP baselines in both fully unsupervised and supervised settings, and can generalize to other sessions without additional distillation while maintaining superior performance. These findings demonstrate that cross-modal knowledge distillation is a powerful and scalable approach for leveraging high-performing spike models to develop more accurate LFP models.

## 1 Introduction

Recent advances in neural recording technologies have enabled the collection of large-scale neural datasets across multiple subjects and recording sessions and led to many advanced models of neural activity that are trained for single recording sessions [1–12]. Moreover, access to such large-scale datasets has motivated training multi-subject and multi-session models of neural activity that can generalize across experimental conditions and tasks [13–22]. To date, the development of multi-session models has focused on spiking activity given the widespread availability of spike datasets compared to other neural modalities such as field potentials. Indeed, local field potentials (LFPs) remain underutilized in recent modeling efforts, despite being routinely recorded alongside spikes. Yet, LFP signals can offer a complementary modality for neuroscience investigations by measuring the brain at larger spatiotemporal scales compared with neuronal spikes [23, 24]. Furthermore, LFPs are potentially advantageous for brain-computer interfaces (BCIs) [25–27] for several reasons. First,

---

*Corresponding author: shanechi@usc.edu

39th Conference on Neural Information Processing Systems (NeurIPS 2025).

they often exhibit greater long-term stability and robustness as they are less sensitive to small shifts in electrode positions or neuronal loss due to reflecting larger-scale population activity across many neurons [23, 24, 28–30]. Second, they are more consistently available than spikes, particularly in chronic long-term settings where spike recordings often degrade or become unavailable over time [28, 31–34]. Third, they have lower power requirements than spike signals, making them more suitable for real-time applications such as BCIs [29, 35].

Despite these advantages, models trained on LFP signals tend to underperform compared to spike-based models in decoding tasks, especially under unsupervised and self-supervised training regimes [9, 24, 34, 36, 37], which are critical for modeling with unlabeled neural data and for building neural representations that are generalizable across tasks. This gap arises primarily due to inherent challenges in modeling LFP signals. Specifically, LFP signals reflect population-level aggregate activity from complex neural circuits, resulting in difficulties in isolating individual sources of task-relevant neural variability within aggregate signals [38, 39], and leading to redundant and highly correlated spatial patterns [40]. In addition, high noise correlations in low-frequency bands of LFP [41] can further obscure task-relevant information in LFP signals.

To overcome these limitations and enable training more accurate LFP models, we propose a **cross-modal representational knowledge distillation framework** that leverages spike-based transformer models pretrained on an abundant amount of public spike datasets as teacher models to improve the quality of LFP transformer models. We enable our framework to operate in a fully unsupervised setup, which is of primary interest, while also extending it to the supervised scenario. We demonstrate that representational knowledge distillation from pretrained models of high-fidelity spikes guides the LFP models toward capturing behavior-predictive features, thus remarkably improving the decoding performance of LFP models while mitigating the effects of redundant and noisy components typically observed in unsupervised LFP training and preserving the generalization properties of LFP signals.

**Contributions** In summary, we make the following contributions: 1) We develop a novel unsupervised cross-modal knowledge distillation framework to transfer representational knowledge from pretrained multi-session spike teacher models to LFP-based student models. Through extensive empirical evaluation across motor cortical data from 6 monkeys that span 3 distinct datasets with LFP and LFP power signals, we demonstrate that distillation significantly improves the downstream decoding accuracy of LFP models while also maintaining their generalizability to other sessions that are not used for distillation. The distillation gains persist even in an extended supervised setting. 2) We develop an extension that performs multi-session cross-modal knowledge distillation and show that it can further improve downstream decoding performance. 3) We also develop a multi-session LFP-only baseline model (MS-LFP) trained on these motor cortical datasets, enabling rigorous evaluation and comparison of LFP models. 4) As an additional contribution and to build our teacher models, we develop an improved multi-session spike (MS-Spike) model by designing new neural tokenization strategies and training optimizations, and demonstrate that it outperforms a recent state-of-the-art baseline (NDT2 [17]) on downstream behavior decoding tasks. Taken together, our results highlight cross-modal representational knowledge distillation as a powerful strategy to leverage complementary neural modalities both for investigating cross-modal neural representations and for developing robust and scalable neural decoding models for applications such as BCIs.

## 2 Related Work

**Neural data modeling** Many approaches in neural data modeling primarily focus on training models within individual recording sessions, particularly on spiking signals and using deep learning-based approaches [1–4, 6–9, 11, 12, 42], or state-space models [5, 10, 43]. More recently, multi-session modeling strategies have been introduced to improve generalization across sessions and subjects, employing techniques such as session-stitching [2], contrastive-learning [13], and transformer-based approaches with different neural tokenization strategies using either unsupervised [16, 17] or supervised [14, 15, 44] objectives. To develop our spike teacher models, inspired by these works, we adopted a modified neural tokenization approach that led to an improved multi-session spike model, which we then used as the teacher network for our novel cross-modal knowledge distillation framework. Beyond spike modeling, recent efforts have developed multi-session modeling frameworks for intracranial electroencephalography (iEEG) signals by applying temporal patch-based tokenization, in some cases combined with spatial embedding of electrode locations, and trained these models using a masked autoencoding (MAE) objective [18, 19], self-supervised methods [20],

or supervised objectives [21]. However, intracortical LFP signals are distinct from iEEG signals, which cover a larger spatial scale of neural activity than LFP due to distinct electrode types. For our LFP modalities, we design a session-specific spatial patch tokenization. Given the continuous nature of LFP recordings and intracranial recordings, we also train our multi-session LFP models under the MAE objective.

**Multimodal neural modeling**   The goal of our cross-modal knowledge distillation is distinct from prior works that model multimodal neural signals to enable information fusion during inference, for example, using autoencoder-based architectures [45, 46] and state-space models [24, 30, 34, 47–51]. Unlike our framework, the goal of these methods is to collectively model both modalities for information fusion and, as such, they require access to both modalities at inference time. In contrast, our goal is to develop an improved unimodal LFP model that operates *solely* on LFP signals after training and during inference. Developing accurate unimodal LFP models is critical both to study larger-scale circuit-level neural representations and because LFPs are the only available modality in many recording scenarios (see Introduction). While our goal is distinct from multimodal decoding, to more comprehensively evaluate our framework, we also developed and compared it with a multimodal baseline model of spike and LFP signals, and performed inference/decoding either by processing both modalities or only unimodal LFPs.

**Knowledge distillation**   Outside neuroscience, knowledge distillation has been widely adopted in deep learning applications to transfer the learned structure from an often larger teacher model to a smaller student model [52], with successful application across various domains such as natural language processing [53, 54] and computer vision [55, 56] through classification-level logit matching or intermediary feature-level alignment objectives. While most prior works focused on distilling the knowledge of a larger model to a smaller model of the same modality, in domains outside neuroscience, knowledge distillation across different modalities has also been studied through alignment-based strategies [57, 58], including contrastive distillation objectives [59]. In neuroscience, knowledge distillation has recently been used for the distinct goal of learning under privileged information to improve behavior decoding from spiking activity by distilling knowledge from a teacher model that observes behavior as privileged knowledge in addition to spikes [60]. However, cross-modal knowledge distillation between neural modalities remains unexplored. Furthermore, it is unclear how such knowledge distillation can affect downstream behavior decoding. Here, we develop a novel cross-modal representational knowledge distillation framework for neural modalities and show its substantial benefit for learning LFP representations by transferring knowledge from spike models.

## 3   Methodology

In our setup, we have access to $M$ sessions of neuronal recordings, all containing discrete spiking activity. Among these $M$ sessions of available recordings, only a subset $L \ll M$ sessions contain continuous LFP signals (see Table 2). Furthermore, each recording session can differ substantially, including variations in the number of recorded spiking neurons, LFP channels, behavioral variables, and recording durations. Also, in many scenarios, neural data may be unlabeled in some sessions due to a lack/difficulty of behavioral measurement or annotation, thus requiring an unsupervised approach. We develop an unsupervised cross-modal knowledge distillation framework that transfers knowledge about latent neural representations from multi-session spike models into LFP models. We also develop a supervised extension of our framework to show the robustness of our findings.

### 3.1   Neural signal tokenization

The first step toward our cross-modal knowledge distillation framework involves utilizing the available large-scale spike recordings to build a robust and generalizable multi-session, multi-subject **unsupervised** model for spiking activity as the teacher model. This model should capture key neuronal dynamics across various experimental tasks and recording setups, thereby facilitating effective knowledge transfer to LFP signals. To achieve that, we develop a multi-session spike transformer encoder model. We do so by adopting and modifying the approach in [17], namely NDT2, which uses shared space embeddings across sessions, while accounting for cross-session/subject spatial variability via learned session/subject tokens. Different from [17], we encode spatial variability across sessions via session-specific space embeddings, offering a direct spatial parameterization as detailed below.

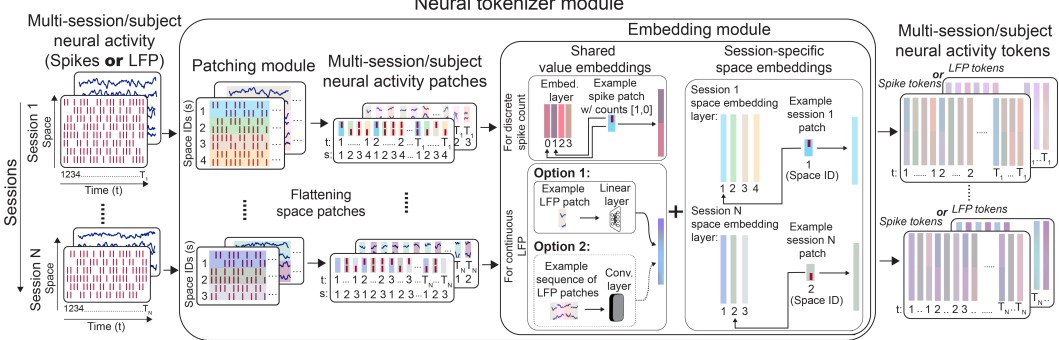

Figure 1: Tokenization of multi-session spiking activity or LFP signals. First, neural signals are patched across space (i.e., neurons or channels) with padding if necessary. Then, the embedding module learns value embeddings for neural signals using the same block that is shared across sessions, as well as session-specific space embeddings for each spatial patch. The final token representations are formed by adding the value embeddings (shared across sessions) and session-specific space embeddings. This addition is illustrated by mixing the colors of value and space embeddings.

As shown in Fig. 1, the first step is to form token sequences out of multi-session neural recordings. Assume that we have spike count recordings $\boldsymbol{s}_t^i \in \mathbb{N}_0^{n_s^i}$ where time-index $t \in \{1, 2, \ldots, T_i\}$, session-index $i \in \{1, 2, \ldots, M\}$ and $n_s^i$ is the number of recorded neurons for session $i$. As done in NDT2, inspired by image-like patching of time-series signals [61] across time, for each recording session $i$, we first group (patch) $S$ neurons across space (with padding if $n_s^i$ is not divisible by $S$). Each neural patch is later processed as a separate token, and $S$ is a hyperparameter denoting the spatial patch size. Thus, for each $t$, the tokenizer module generates $\lceil \frac{n_s^i}{S} \rceil$ neural patches, which increases the total number of tokens (i.e., the sequence length processed by the transformer encoder) from $T_i$ to $T_i \lceil \frac{n_s^i}{S} \rceil$. This **strictly spatial** patching approach allows us to 1) maintain the temporal resolution of the original neural activity and 2) enable the encoder backbone to explicitly leverage spatial relationships (e.g., via attention mechanism) in addition to temporal patterns.

Once patches are formed, for each neural patch, we first learn value embeddings using the same block shared across all recording sessions. For discrete spike counts, we learn embedding vectors for each unique spike count value, $V = \{V_0, V_1, \ldots, V_k\}$ where $k$ denotes the maximum spike count allowed (e.g., 5 spikes in 10 ms bins), each $V_i \in \mathbb{R}^{\frac{d}{S}}$ and $d$ denotes the final token dimensionality (or latent dimensionality). Thus, we form the value embedding of each patch, denoted by $E_v \in \mathbb{R}^d$, by concatenating the value embedding vectors of each neuron's spike count (i.e., $V_i$'s) within that patch.

To account for spatial variability both within and across sessions, we learn session-specific space embedding vectors for each neural patch in every session, $E_{sp}^i = \{E_1^i, \ldots, E_{\lceil \frac{n_s^i}{S} \rceil}^i\}$ where each $E_j^i \in \mathbb{R}^d$. This differs from NDT2, which accounts for spatial differences across sessions by using session- and subject-specific embedding vectors, while space embeddings are shared across sessions (see Appendix A.6 for performance comparisons of our multi-session spike model with NDT2 and details of the NDT2 baseline). Finally, we form the token embedding sequence by adding the value and space embedding vectors $E_v$ and $E_j^i$ for each neural patch.

In addition to the multi-session spike model (MS-Spike), we also developed a multi-session model for LFP signals (MS-LFP). This allows us to assess the benefit of distilling knowledge from multi-session spike models into LFP models more rigorously. To do so, we extended our patch-based tokenization strategy for LFP signals. However, since LFP signals are continuous-valued, we formed the value embeddings either 1) by passing each neural patch through a shared linear layer or 2) by passing the sequence of neural patches through a dilated causal convolutional layer.

## 3.2 Pretraining and fine-tuning of multi-session models

After the tokenization, we **pretrain** the multi-session models in an **unsupervised** manner, whether for spikes or LFP, using an unsupervised masked autoencoding (MAE) objective as shown in Fig. 7. We drop 60% of tokens across space and time, and train the transformer encoder and predictor to

reconstruct the neural activity from the unmasked tokens. Then, we use the pretrained tokenizer and transformer encoder to extract latent representations.

Once pretraining of multi-session models is complete, we **fine-tune** on individual recording sessions in an unsupervised manner with the same MAE objective (**unsupervised setting**). For an unseen session, we initialize new spatial embeddings for its neural patches and optimize them during fine-tuning. While pretraining of multi-session models is always unsupervised, to show the robustness of our findings, we also explore performing the fine-tuning by further incorporating a supervised behavior regression objective. To do so, a new behavior regression head is coupled to the encoder backbone. For the **supervised variant**, we jointly optimize MAE and behavior regression losses, whereas only behavior regression loss is optimized for the **fully-supervised variant**. The above gives us the MS-Spike, which we can now utilize to develop our cross-modal knowledge distillation framework. Please also see Appendix A.1 for additional details on pretraining and fine-tuning.

### 3.3 Cross-modal knowledge distillation

After training robust and generalizable models of multi-session spiking activity, we transfer knowledge from these high-performance spike models to a model operating solely on LFP signals. We perform the knowledge distillation in the latent representational space to leverage the representational power of pretrained spike models. Specifically, we propose a cross-modal knowledge distillation framework that aligns representations across spiking activity and LFP signals as depicted in Fig. 2.

The first step in this approach is fine-tuning the pretrained multi-session spike model by using one of the fine-tuning approaches defined in Section 3.2, either fully unsupervised or with some level of behavior supervision if desired. Then, we initialize a new model for the single-session LFP signal with the patch tokenization strategy described in Section 3.1 and shown in Fig. 1. Unlike spike signals, we use a dilated convolutional layer for the value embeddings of LFP signals instead of learnable embeddings.

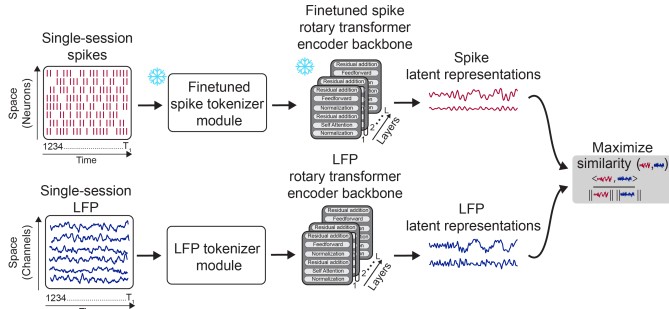

Figure 2: Cross-modal knowledge distillation across spiking activity and LFP signals via representation-level alignment.

Next, to train the LFP model using distillation, we align the latent representations of paired spike and LFP signals—after averaging the representations of spatial patches corresponding to the same timestep—by maximizing their average cosine similarity. To prevent overfitting to the alignment objective and also allow for inclusion of LFP-specific dynamics, we include an additional autoencoding loss to reconstruct the observed LFP signals. The final distillation objective combines both components:

$$\mathcal{L}_{distill} = \underbrace{\frac{1}{n_y T} \sum_{t=1}^{T} ||\boldsymbol{y}_t - f_\phi(f_\theta(\boldsymbol{y}_t))||_2^2}_{\text{Autoencoding objective}} + \lambda \underbrace{\left( 1 - \frac{1}{T} \sum_{t=1}^{T} \frac{< f_\theta(\boldsymbol{y}_t), f_\gamma(\boldsymbol{s}_t) >}{||f_\theta(\boldsymbol{y}_t)||_2 ||f_\gamma(\boldsymbol{s}_t)||_2} \right)}_{\text{Representation-level alignment/similarity objective}} \quad (1)$$

where $\lambda$ is the scaling hyperparameter, $\boldsymbol{y}_t \in \mathbb{R}^{n_y}$ and $\boldsymbol{s}_t \in \mathbb{R}^{n_s}$ denote paired LFP and spike signals, respectively, and $< \cdot, \cdot >$ denotes inner product. The function $f_\phi(\cdot)$ is a predictor linear layer used to reconstruct $\boldsymbol{y}_t$, while $f_\theta(\cdot)$ and $f_\gamma(\cdot)$ represent the tokenizer module and transformer encoder backbone stacks for LFP and spike modalities, respectively. Note that the spike model $f_\gamma(\cdot)$ is kept **frozen** when optimizing the distillation objective, serving as the teacher in the distillation process (see Appendix A.8.2 for the ablation study on the systematic unfreezing of the teacher MS-Spike model).

Overall, the required steps for the distillation procedure can be summarized as 1) pretraining an MS-Spike model on pretraining sessions, 2) fine-tuning the MS-Spike model on a new session using one of the fine-tuning tasks, depending on the level of supervision, and 3) initializing a single-session

LFP model, then training it on the new session in step 2 via the cross-modal knowledge distillation objective in Eq. 1 where the teacher model is the fine-tuned MS-Spike model from step 2. During this step, the teacher MS-Spike model is fully frozen.

# 4 Results

We first pretrained a multi-session spike model (**MS-Spike**) on 226 sessions collected from 16 subjects across 6 different datasets [62–72] (Table 2). To test generalizability to unseen sessions, during pretraining, we held out 5 sessions from [62, 63] and 3 sessions from [64] for fine-tuning, which were amongst sessions with paired spike-LFP recordings (standard low-pass filtering was used to extract raw LFP signals in these datasets, see Appendix A.3.1 for details). Additionally, to test generalizability to unseen subjects and datasets, during pretraining, we held out all sessions from [73], which contained paired spike and LFP-power signals (standard power band features were provided in the dataset). After pretraining, we fine-tuned the MS-Spike model on the spikes of the held-out sessions and then trained Distilled LFP models using the fine-tuned spike models as teachers.

As our primary comparisons, we compared the performance of knowledge Distilled LFP models with the following transformer models, which had the same architecture as described in Section 3.1:

- **Multi-session LFP models (MS-LFP)**: These models were pretrained on LFP signals of recording sessions from [62, 63] and [64] (25 and 9 sessions, respectively), and then fine-tuned on the LFP signals of sessions that were held out of pretraining (same as the 5 and 3 held-out sessions from MS-Spike pretraining). Because [73] provides LFP-power signals rather than raw LFP signals, we pretrained separate multi-session models on these LFP power signals from 14 sessions across 3 subjects and fine-tuned on 9 held-out sessions from 2 subjects, one of whom was not included in the pretraining data.

- **Single-session LFP models (SS-LFP)**: These models were trained independently on the LFP (or LFP power for [73]) signals from each individual recording session.

- **Single-session multimodal models (SS-MM)**: These models were trained on the concatenation of spike and LFP (or LFP power) signals from individual sessions, and used both modalities during training and inference.

- **Single-session multimodal models with zero spikes during inference (SS-MM-ZS)**: To directly compare with LFP-only models at inference time, we also evaluated the SS-MM models by zeroing out the spike inputs to the tokenizer (mean-imputation after z-scoring), thus having the model process only LFPs while keeping the architecture unchanged.

All models were trained following the setup described in Section 3.1 and Appendix A.1.

## 4.1 Cross-modal knowledge distillation significantly improves behavior decoding

As our primary comparisons, we compared the behavior decoding performance of the Distilled LFP models in the fully unsupervised setting, where MS-Spike was fine-tuned only with the unsupervised MAE objective and was used as teacher for the Distilled LFP models with the objective in Eq. 1. All baseline LFP models are also trained (and fine-tuned for MS-LFP) to optimize the MAE objective with a linear layer for value embeddings as described above.

We found that distillation leads to a remarkable improvement in decoding performance of the LFP signals as shown in Fig. 3. Distilled LFP models (orange line) consistently outperformed the models trained solely on the LFP signals, whether using multi-session (MS-LFP) or single-session (SS-LFP) training ($p < 2.6 \times 10^{-10}$ for both MS-LFP and SS-LFP, $n = 51$, one-sided Wilcoxon signed-rank test). For example, for Monkeys I, C, Ch, and M, respectively, the average behavior decoding performance ($R^2$) of the Distilled LFP model was 0.66, 0.72, 0.77, and 0.72 (0.71 on average), whereas the MS-LFP model achieved 0.24, 0.34, 0.33, and 0.22 (0.27 on average), and the SS-LFP models underperformed the MS-LFP model (0.24 on average).

Also interestingly, we observed that Distilled LFP models performed similarly to or sometimes better than their teacher MS-Spike models, with the Distilled LFP average performance being higher than that of MS-Spike models (0.71 vs. 0.69 on average; $p < 1.7 \times 10^{-3}$, $n = 51$, one-sided Wilcoxon signed-rank test). Since shared information between the spike and LFP modalities has been shown to

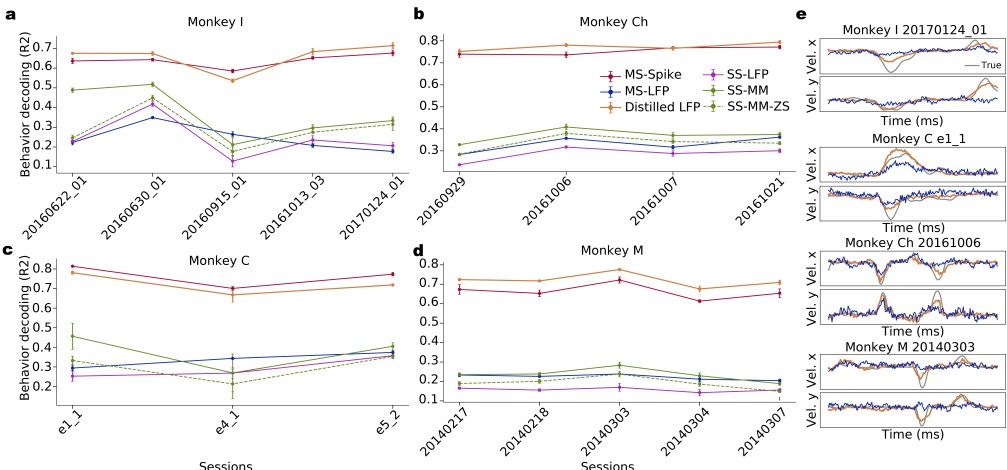

Figure 3: Behavior decoding performances ($R^2$) of unsupervised models. (a-d) Decoding results across sessions for individual monkeys (Monkeys I, C with LFP signals, and Monkeys Ch, M with LFP power signals). Error bars around each datapoint indicate standard deviations obtained across 3 different seeds. (e) The true (grey) vs. decoded behavior trajectories (velocities) from the latent representations of Distilled LFP models (orange) and MS-LFP models (blue).

be task-predictive in prior works [23, 24, 74], this result suggests that Distilled LFP models potentially learn to better extract shared behavior-predictive information compared with MS-Spike models due to aligning the LFP representations with their associated spike representations.

To further explore the power of knowledge distillation in extracting informative embeddings from LFPs, we trained multimodal SS-MM models and performed inference either with multimodal signals (SS-MM, solid green line), or only with LFP signals by zeroing out the spike signals, i.e., imputing with their global-mean after z-scoring (SS-MM-ZS, dashed green line). We found that SS-MM and SS-MM-ZS achieved 0.33 and 0.27 average decoding performance, respectively, which were again substantially lower than the Distilled LFP models ($p < 2.6 \times 10^{-10}$, for both SS-MM and SS-MM-ZS $n = 51$, one-sided Wilcoxon signed-rank test). This result indicates that in the fully unsupervised setting, distillation is more successful in learning informative embeddings for LFP signals compared with the concatenation of spike and LFP modalities in the input to the transformer, which presents a distinct difficulty of learning all collective dynamics in both signals. Indeed, the distillation objective provides a principled and scalable approach to unsupervised representation learning by explicitly aligning the representations across modalities.

In addition to the held-out sessions reported above, we also performed the same analysis for the sessions used in the pretraining of MS-Spike and MS-LFP models (25, 9, and 14 sessions for [62, 63], [64] and [73], respectively) and found that the Distilled LFP models again improve behavior decoding performance over baseline models for the pretraining sessions (see Fig. 11 in Appendix A.10).

To assess the scalability of distillation, we trained Distilled LFP models with different model sizes. Fig. 10 shows that even when the Distilled LFP model capacity was decreased 10 times compared to the other LFP-based baselines, the distilled models outperformed all baselines regardless of scale.

Further, in an ablation analysis on the impact of MS-Spike pretraining dataset size (Appendix A.8.3), we found that the performance difference between the MS-Spike and MS-LFP models is not simply due to the dataset size used during pretraining but also to the differences in signal characteristics. Importantly, we found that this gap can be overcome through our cross-modal knowledge distillation.

## 4.2 Cross-modal knowledge distillation successfully aligns LFP and spike representations

Next, we investigated the alignment of latent representations extracted by MS-Spike, MS-LFP, and Distilled LFP models by computing 2-dimensional t-SNE representations from the sequence-averaged latent representations. As shown in Fig. 4, the latent representations of MS-Spike and Distilled LFP models are clustered together, indicating their successful alignment through the distillation objective, whereas the latent representations of MS-LFP models are clustered far away from both. Also, for all

models, the latent representations of the sessions from the same subject are clustered closer (shown by gray ovals), despite not having explicit learnable subject tokens.

We also quantified the degree of alignment across models at the sequence-level by computing the top-1 and top-5 representation retrieval accuracy and the mean rank of the paired spike and LFP sequence representations among all (~3000) sequences. We also computed the alignment at the global representation level by computing the centered kernel alignment (CKA) [75] between the MS-Spike, MS-LFP, and Distilled LFP models (see Appendix A.5 for details). We observed that MS-Spike and Distilled LFP representations exhibit a high degree of alignment, achieving 0.67 top-1 accuracy, 0.90 top-5 accuracy, a low mean rank of 4.99, and a CKA of 0.89. In contrast, MS-Spike vs. MS-LFP and Distilled LFP vs. MS-LFP pairs yield much worse retrieval performance in all metrics, as seen in Table 1. Even though the sequence-level retrieval metrics for MS-Spike vs. MS-LFP and Distilled LFP vs. MS-LFP are close to the random baseline, their global-level CKA values (0.70 and 0.76) are relatively high—though much

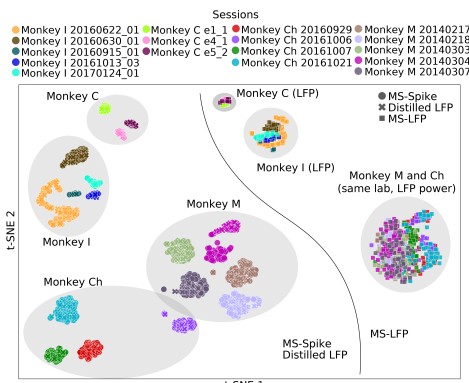

Figure 4: t-SNE representations of time-averaged latent representations across MS-Spike, MS-LFP, and Distilled LFP models across recording sessions from 4 subjects held-out for fine-tuning.

lower than that of MS-Spike vs. Distilled LFP (0.89)—suggesting global-level similarities between the spike and LFP representation spaces (see Appendix A.5 for details). These results indicate that, in addition to improving the quality of LFP models, the cross-modal knowledge distillation framework can also serve as a new tool for aligning and investigating latent structure across different neural modalities at multiple spatiotemporal scales.

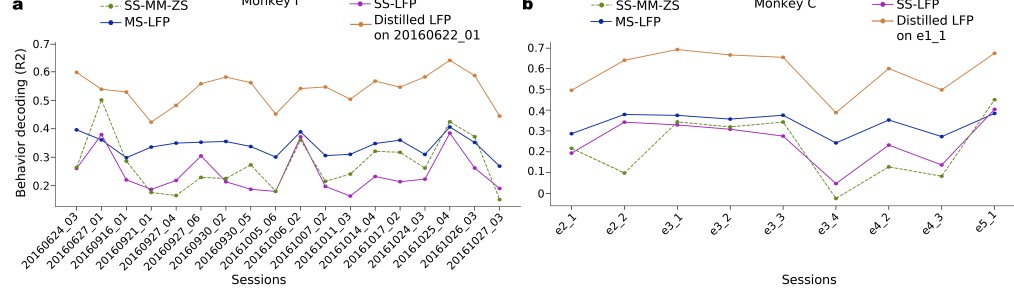

Figure 5: Behavior decoding performances ($R^2$) of Distilled LFP models on LFP signals of the sessions denoted on the x-axis that were never used in pretraining, fine-tuning, or distillation for Monkeys I (a) and C (b).

## 4.3 Distilled LFP models can generalize to other sessions without additional distillation

To evaluate whether Distilled LFP models can generalize beyond the session used for distillation, we tested their decoding performance on LFP signals from sessions that were never used in the MS-Spike pretraining, fine-tuning, or distillation. To this end, we trained another MS-Spike model where all sessions shown in Fig. 5 were held-out during pretraining. We then fine-tuned this MS-Spike model separately on two of the held-out sessions—20160622_01 for Monkey I and e1_1 for Monkey C—and trained Distilled LFP models on the LFP signals of these sessions using their corresponding fine-tuned MS-Spike models as teachers. Finally, we froze these Distilled LFP models and evaluated them on the LFP signals of the remaining held-out sessions to assess their generalization performance.

As shown in Fig. 5, Distilled LFP models trained only on a single session's spike–LFP alignment (e.g., session 20160622_01 for Monkey I in panel a and session e1_1 for Monkey C in panel b) substantially outperform all other LFP baselines, including MS-LFP, SS-LFP, and SS-MM-ZS, which were trained (or fine-tuned, in the case of MS-LFP) on the tested sessions' LFP signals. This result suggests that the distillation objective enables the transfer of the teacher MS-Spike model's prior

knowledge from its pretraining distribution into the Distilled LFP models, allowing them to leverage spike-aligned representations even on the held-out tested sessions, despite no spike or LFP data from those sessions being used during distillation, which was performed on a different held-out session.

## 4.4 Cross-modal knowledge distillation improves performance even in the supervised fine-tuning setting

Our primary goal is to develop a fully unsupervised distillation framework, motivated by the broader generalization capabilities of unsupervised models and/or the sparsity of behavior measurements or annotations in many datasets. Nevertheless, we also examined whether cross-modal distillation improves performance even in the supervised case. In this case, multi-session MS-Spike and MS-LFP models are fine-tuned with a supervised variant, where we jointly optimize the MAE and behavior regression objectives. Also, all single-session models are trained with the behavior regression objective. We used a convolutional layer for value embeddings for all LFP models. The Distilled LFP model was trained with the objective in Eq. 1 using the above supervised teacher spike models.

As shown in Fig. 6, Distilled LFP models again consistently outperformed all LFP-only baselines, including MS-LFP, SS-LFP, and SS-MM-ZS, in behavior decoding performance ($R^2$). As expected, unlike in the unsupervised setting, the Distilled LFP models did not surpass the spike-based models (MS-Spike and SS-MM), which served as an upper bound on the decoding performance. This is likely due to the direct supervision of spike models with behavior, which helps the spike model uncover more behavior-predictive embeddings than in the unsupervised case. However, as noted

| Model Pairs | Top-1 ↑ | Top-5 ↑ | Mean rank ↓ | CKA ↑ |
|---|---|---|---|---|
| MS-Spike Distilled LFP | 0.6711 | 0.9041 | 4.99 | 0.89 |
| MS-Spike MS-LFP | 0.0007 | 0.0023 | 1289.15 | 0.70 |
| Distilled LFP MS-LFP | 0.0003 | 0.0023 | 1265.57 | 0.76 |
| Random | 0.0003 | 0.0013 | 1513.42 | 0.07 |

Table 1: Representation retrieval metrics across pairs of MS-Spike, MS-LFP, and Distilled LFP models. The last row demonstrates the metrics computed across random representations.

above, the Distilled LFP models significantly outperformed all LFP-only baselines ($p < 2.6 \times 10^{-10}$ for MS-LFP, $p < 2.9 \times 10^{-10}$ for SS-LFP, and $p < 2.6 \times 10^{-10}$ for SS-MM-ZS, $n = 51$, one-sided Wilcoxon signed-rank test). These results show that the distillation framework benefits LFP modeling even when supervision with behavior is performed for fine-tuning, leading to powerful LFP models.

We also tested the distillation performance in this supervised setting for the recording sessions used in pretraining of MS-Spike and MS-LFP models. We found that the Distilled LFP models again improved downstream decoding performance over all LFP-only baselines (see Fig. 12 in Appendix A.10). As an additional analysis, we also trained Distilled LFP models in a fully-supervised setting by replacing the autoencoding objective with the behavior regression objective in Eq. 1 and by fine-tuning the teacher MS-spike models fully-supervised. As shown in Figs. 13 and 14, fully-supervised distillation can further improve downstream decoding performance compared to baseline methods.

## 4.5 Multi-session cross-modal knowledge distillation further improves performance

Next, we trained the distilled LFP models in a multi-session setup (MS-Distilled LFP) to investigate whether multi-session distillation further improves downstream decoding performance over single-session distillation. In this setup, the fully unsupervised distillation objective in Eq. 1 was optimized on pooled spike-LFP data across multiple sessions rather than on just the target session, and the distilled model was then fine-tuned on the behavior-labeled target session (see Appendix A.12). This setting is particularly important in practical scenarios where large amounts of unlabelled paired spike-LFP data are available across multiple sessions or subjects, but behavior labels are limited to a small subset of sessions. Through this approach, we aim to leverage shared structure across sessions to obtain more accurate and generalizable initialization even beyond single-session distillation.

Extending to multi-session distillation provided robust gains over single-session Distilled LFP models. MS-Distilled LFP models significantly outperformed Distilled LFP models for supervised fine-tuning (0.82 vs. 0.80, $p < 1.3 \times 10^{-4}$, $n = 51$, one-sided Wilcoxon signed-rank test), and fully-supervised fine-tuning (0.83 vs. 0.81, $p < 1.3 \times 10^{-6}$, $n = 51$, one-sided Wilcoxon signed-rank test) strategies as shown in Figs. 16, 17, while achieving comparable performance for the unsupervised fine-tuning setting (0.72 vs. 0.71) (Fig. 15). Also, as shown in Figs. 18, 19, 20, similar results held for the

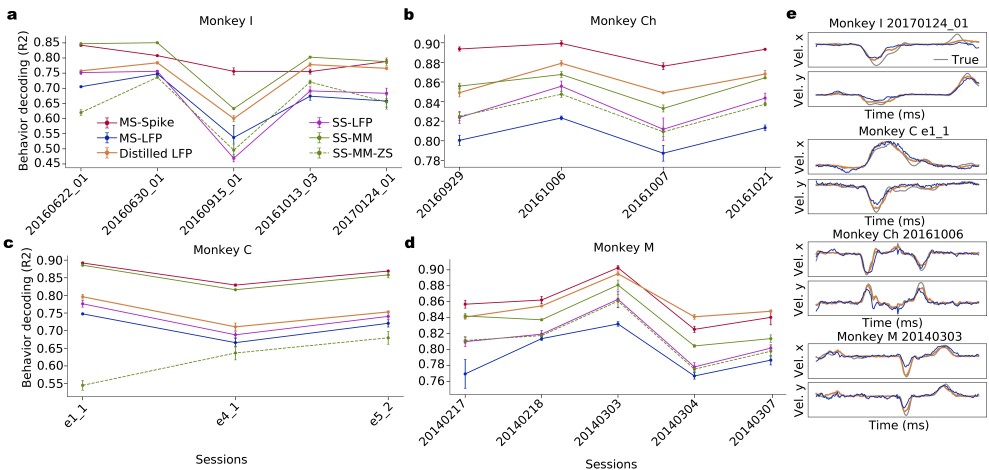

Figure 6: Behavior decoding performances ($R^2$) of supervised models. Figure conventions are the same as in Fig. 3.

recording sessions used in pretraining of MS-Spike, MS-LFP, and MS-Distilled LFP models. Overall, these results highlight that multi-session distillation can leverage *large-scale unlabeled neural data* to build more accurate and generalizable LFP models.

## 5 Discussion

Here, we developed a cross-modal representational knowledge distillation framework to build more accurate LFP models by leveraging high-performing spike-based models as teachers. To achieve that, we first developed an improved multi-session spike model with a new neural tokenization strategy and pretrained it on publicly available motor cortical datasets. Then, we transferred the representational knowledge of spike models to LFP models through a representation-level alignment objective. To enable rigorous evaluations of our approach, we also developed a multi-session LFP-only baseline trained on motor cortical LFP signals. Our results demonstrate that cross-modal distillation significantly improves the representation quality of the LFP models, both in fully unsupervised and supervised settings, even when using distilled models with ten times fewer parameters. In addition, we showed that cross-modal distillation can be an important scientific tool for aligning and investigating latent structure across multiple spatiotemporal scales of brain activity. Moreover, we showed that the distilled models can generalize to LFP signals from other sessions without additional distillation, still achieving superior decoding performance compared to LFP baselines. Finally, we showed that extending the distillation approach to a multi-session setting can further improve decoding performance.

There are several areas for future investigation. First, while we incorporated substantial pretraining data, extending pretraining to incorporate larger and more diverse multi-session datasets may reveal even richer latent representations across sessions, ultimately enabling more diverse evaluations and improving generalization and robustness. Second, investigating more diverse pretraining tasks beyond masked autoencoding is an important future direction to potentially improve the quality of the distilled models. Third, this distillation approach can be tested for knowledge transfer beyond cross-modal transfer, e.g., for transfer between different brain regions. Moreover, multimodal models represent a promising complementary direction when both modalities are available at inference time, as they can leverage information from both spike and LFP signals to potentially surpass unimodal performance. While we compared to a multimodal approach as a baseline, our simple proof-of-concept multimodal baseline could be extended into more sophisticated architectures in the future for better multimodal performance. Nevertheless, as spike signals degrade faster than LFP signals in BCIs (e.g., due to electrode tip encapsulation) [28, 76–78], the Distilled LFP models provides a powerful approach to maintain high BCI performance over long time-periods, given the higher stability of LFPs, thus improving the robustness and longevity of BCIs. Finally, while our results highlight the potential of LFP-based models as robust and accessible alternatives to spike-based models, confirming their reliability as full replacements requires further experimental validation under controlled signal degradation conditions.

## Acknowledgements

This work was partly supported by the National Institutes of Health (NIH) grants RF1DA056402, R01MH123770, and R61MH135407, the NIH Director's New Innovator Award DP2-MH126378, and the National Science Foundation (NSF) CRCNS program award IIS-2113271. We sincerely thank Danil Tyulmankov and Enes Burak Bilgin for their feedback, and all members of the Shanechi lab for helpful discussions.

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

# A   Appendix

## A.1   Details of pretraining and fine-tuning of the multi-session models.

**Pretraining**   To allow for the training of a generalizable and robust foundational model of multi-session neural activity, we employed masked autoencoding (MAE) as the unsupervised training objective due to its success in various applications using transformer architectures [17, 79, 80].

Specifically, we randomly drop $x\%$ (e.g., 60%) of the neural patch tokens (after the tokenizer module) before passing them through the transformer encoder backbone. Note that the token dropping is not only performed across time, but also across space, allowing the model to reconstruct the missing information from both temporal and spatial contexts. Then, we append learnable mask tokens to the token sequence at the positions of the dropped patches and add their corresponding space embeddings to the mask tokens to inform the model of their spatial identity. The resulting token sequence is then passed through a predictor transformer, and the output representations at the masked positions are treated as reconstructions of the dropped patches. Both the encoder and predictor use transformer architectures with rotary positional embeddings [81] (see the ablation study in Appendix A.8.1 for details about our choice of using rotary transformers as the encoder backbone architecture). As the training objective, we employed a Poisson negative log-likelihood (NLL) for discrete spiking activity and mean-squared error (MSE) for continuous LFPs. During loss computation, any padded dimensions introduced during spatial patching are excluded. Further, to train multi-session (MS) distilled LFP models, we optimized the training objective in Eq. 1 across multiple recording sessions rather than on individual recording sessions (see Appendix A.12 for details).

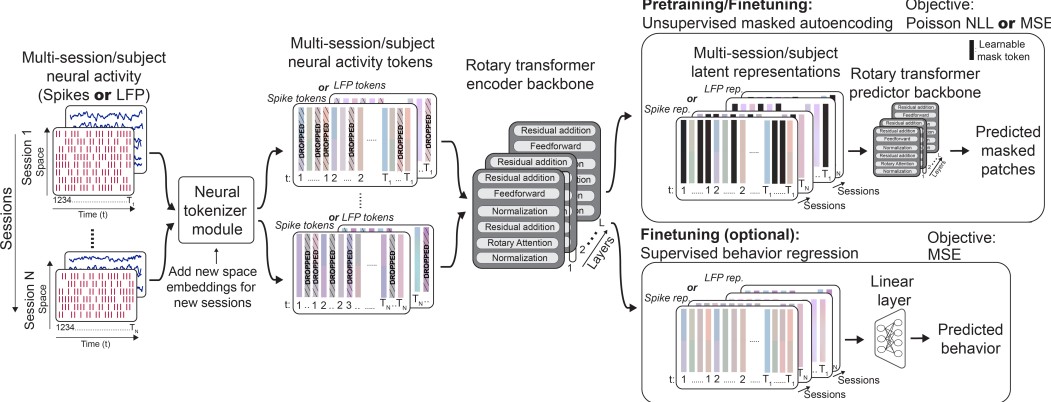

Figure 7: Overall architecture of the proposed multi-session model. As the self-supervised pretraining objective, we employ masked autoencoding (MAE) to allow for learning generalizable representations across datasets recorded during diverse experimental tasks. During fine-tuning on a single session, we optionally incorporate supervised behavior regression. MSE: mean-squared error. NLL: negative log-likelihood. Rep.: Representation.

**Fine-tuning**   After pretraining the multi-session model with the unsupervised MAE objective, we primarily fine-tune the model in an **unsupervised setting**. This approach reflects our core goal: to build general-purpose neural representations that do not rely on task-specific supervision and remain more adaptable across recording conditions and downstream tasks. In addition to this primary unsupervised setting, we also explore two supervised fine-tuning strategies to assess the utility of learned representations. In the **supervised variant**, we jointly optimize MAE and behavior regression losses to align latent representations with behavior while maintaining reconstruction ability. In the **fully-supervised variant**, we fine-tune the model using only the behavior regression objective. All fine-tunings are performed separately on individual single-session recordings. For a new session that is not encountered during pretraining, we initialize new space embeddings for its neural patches and optimize them during fine-tuning. Fine-tuning of the MS-Distilled LFP models was performed the same way as training single-session Distilled LFP models, where the model weights were initialized with the weights of pretrained MS-Distilled LFP model instead of random initialization (see Appendix

A.12 for details). Note that all model parameters are updated during fine-tuning, regardless of the level of behavior supervision.

## A.2 Training details, hyperparameters, and codebase

All models were trained in a cluster with 8 NVIDIA RTX A6000 GPUs. We used automatic mixed precision (AMP) training and flash-attention mechanism [82] in transformer architectures for speed-up and memory efficiency purposes. Also, we implemented our framework with variable-length sequence support to remove any computation and memory overhead that could be caused by sequence padding.

To train multi-session spike models (MS-Spike), we performed distributed training over 4 GPUs with a batch size of 64 with early-stopping patience of 50 epochs, which approximately took 50 hours (~550 epochs), but convergence started around 10 hours of training. We used a patch size ($S$) of 64 for spatial patching of neurons across space. Overall, the MS-Spike model was trained on spike signals from 226 recording sessions, which approximately corresponds to 120.1M tokens. After pretraining, it was fine-tuned on 17 individual recording sessions, comprising roughly 4.8M tokens.

Similarly, for multi-session LFP models (MS-LFP), we performed distributed training over 2 GPUs with a batch size of 64, with early-stopping patience of 50 epochs, which approximately took 3 hours (~560 and ~480 epochs for models trained on LFP and LFP power signals). We used a patch size ($S$) of 32 and 288 for spatial patching for LFP and LFP power signals, respectively, since LFP power signals of [73] had 9 power bands (including local motor potential) from each recording channel.

Single-session models (SS-LFP and SS-MM) and Distilled LFP models were trained on single GPUs with a batch size of 32 for 400 epochs with early stopping patience of 50 epochs, which again took approximately an hour. For SS-LFP and Distilled LFP models, we used a patch size ($S$) of 32 and 288, as done for MS-LFP models, for LFP and LFP power signals, respectively. We used the same patch size ($S$) as in the Distilled LFP models for training the MS-Distilled LFP models, and trained them on 2 GPUs with a batch size of 64 with early stopping patience of 50 epochs, which resulted in convergence around 500 epochs. For SS-MM models, we used a patch size ($S$) of 96 and 352 (summation of patch sizes for MS-Spike and MS-LFP models) for concatenation of spike and LFP or LFP power signals, respectively. For all LFP models whose latent representations were extracted after training (or fine-tuning) with the MAE objective, we used a linear layer for value embeddings to avoid potential information leakage from temporal and spatial convolutions. In the supervised setting, we trained MS-LFP models with a convolutional layer for the value embeddings under the MAE objective, and then fine-tuned with the behavior regression objective to ensure compatibility with single-session models. We found that using a convolutional layer for value embeddings significantly improved the downstream decoding performance for distilled models and supervised single-session models. Therefore, to enable fair comparisons, we selected the best-performing architecture for each model class.

After the pre-training of multi-session models (MS-Spike, MS-LFP, and MS-Distilled LFP), we fine-tuned them for 400 (or 150 for MS-Distilled LFP) epochs, with early stopping patience of 50 epochs, which usually took around an hour on a single GPU.

For all models, we employed 10-layer transformer encoder backbones (except the scaling analysis in Fig. 10) and a hidden state dimension of 256. We used a sequential learning rate composed of 1) a linear warmup learning rate with a start factor of 0.3 that reached its maximum value of 0.000625 over 30 epochs, and 2) an exponential learning rate with a decay factor of 0.995. We used AdamW optimizer [83] with weight decay factor starting from 0.1 and reaching to its maximum value of 0.4 over 1000 epochs (which was never achieved for any model). For models trained/fine-tuned on the MAE objective, we used a masking probability of 0.6 that is randomly applied on neural patches across space and time, and used 4-layer transformer predictors with a hidden state dimension of 192, followed by a 64-dimensional down-projection. For supervised fine-tuning (combination of MAE and behavior regression objectives), we did not apply any scaling on the two loss terms, and for the distillation objective, we scaled the representation alignment objective by $\lambda = 5$.

We release models and an inference notebook for reproducibility at `https://github.com/ShanechiLab/CrossModalDistillation`.

## A.3 Dataset details

All datasets used in this study are publicly available datasets as shown in Table 2, whose experimental and preprocessing details are as follows.

| Dataset | Regions | Experimental Tasks | # Subjects | # Sessions (Pre-Ft) | Total duration (h) | # Total Sequences | # Total Neurons |
|---|---|---|---|---|---|---|---|
| *Makin et al., [62, 63] | M1, S1 | RT | 2 | 42-5 | 13.49 | 9664 | 12060 |
| *Flint et al., [64] | M1 | CO | 1 | 9-3 | 2.03 | 1448 | 2344 |
| *Gallego-Carracedo et al., [73] | M1, PMd, S1 (Area 2) | CO | 4 | 0-23 | 8.21 | 9884 | 1201 |
| Perich et al., [65–69] | M1, PMd | RT, CO | 4 | 109-0 | 34.69 | 34615 | 10301 |
| Churchland et al., [70] | M1, PMd | Maze | 2 | 9-0 | 19.33 | 23117 | 1728 |
| Chowdhury et al., [71] | S1 (Area 2) | TRT, CO | 3 | 12-0 | 7.45 | 8334 | 1188 |
| Ma et al., [72] | M1 | CO, ISO, Key | 4 | 45-0 | 8.24 | 9042 | 4007 |

Table 2: Datasets used during pretraining and fine-tuning of multi-session spike model. Pre-Ft denotes the number of sessions used during pretraining and held out for fine-tuning. * represents datasets that have paired LFP (or LFP power for [73]) signals available. RT, CO, TRT, ISO, and Key are abbreviations for random-target, center-out, two-workspace random-target, isometric wrist torque random-target, and key grasping, respectively.

### A.3.1 Multimodal neural datasets

The following three datasets have simultaneously recorded spike and LFP (or LFP power for [73]) signals, coupled with behavior signals. Therefore, we focused on the following datasets for the distillation analyses. For each dataset, we applied a random 80%–20% train–test split at the segment (sequence) level.

**Makin et al., [62, 63]**  In this dataset, two monkeys (Monkeys I and L) performed a 2D target-reaching task by controlling a cursor in a 2D virtual environment. Both monkeys were trained to perform continuous reaches to circular targets arranged in a grid (e.g., 8x8). Even though there was no inter-trial interval between sequential reaches, there existed a 200 ms lockout interval after a target acquisition during which no target could be acquired. After the lockout interval, a new target was randomly drawn from the set of possible targets with replacement. All sessions include neural recordings from the primary motor cortex (M1, 96 channels), and some sessions additionally include recordings from the somatosensory cortex (S1), yielding 192 channels in total.

Monkeys I and L had 37 and 10 recording sessions, respectively, where broadband signals were publicly available for 30 sessions of Monkey I. For these sessions, we extracted local field potential (LFP) signals through standard processing pipeline: 1) notch filtering at harmonics of 60 Hz (line noise in the U.S), 2) high-pass filtering with 0.05 Hz cut-off frequency to remove DC drift, 3) low-pass filtering below 50 Hz for anti-aliasing, 4) common average referencing to remove interchannel correlations and 4) downsampling to 10 milliseconds (ms) resolution (100 Hz). For spike signals across all 47 sessions, we used sorted units, binned them in 10 ms nonoverlapping windows, and discarded the units with average firing rate below 1 Hz. As downstream behavior signals, we used cursor velocity in $x$ and $y$ directions. For continuous behavior signals and LFP signals, we performed z-scoring for each dimension independently.

Apart from these preprocessing steps, we did not perform further processing such as trial alignment to movement onset or temporal shifting between neural and behavioral signals. All time-series data (spikes, LFPs, and behavior) were segmented into 5-second sequences before being fed into the models. As shown in Table 2, we used all 10 recording sessions from Monkey L and 32 sessions from Monkey I to pretrain the MS-Spike models. Of the 32 sessions from Monkey I, 25 included broadband signals, while the remaining 7 did not. For the MS-LFP models, we used the 25 broadband sessions from Monkey I for pretraining. To evaluate generalization, we held out 5 of these broadband sessions from Monkey I as unseen sessions for fine-tuning.

**Flint et al., [64]**  In this dataset, a monkey (Monkey C) performed a 2D center-out reaching task while grasping a two-link manipulandum. Monkey C was trained to perform reaches from a center position to 2-cm square outer targets in an 8-target environment. Each trial of the task started with the illumination of the center target, where the monkey had to hold the manipulandum for a random hold time of 0.5-0.6 seconds. After, the center target disappeared and an outer target was randomly

selected from the pool of possible 8 targets, which signaled the monkey to start the reach. To obtain the reward, the monkey had to reach the outer target within 1.5 seconds and hold the manipulandum at the outer target for a random time of 0.2-0.4 seconds. Then, the monkey returned back to the center target position, and the next trial started. All 12 recording sessions include neural recordings from M1 (96 channels), with LFP signals available at 2 kHz. We used 2D manipulandum velocity in $x$ and $y$ directions as downstream behavior signals.

To obtain paired spike and LFP signals, we followed the same preprocessing pipeline described above for Makin et al., dataset [62, 63], resulting in synchronized time-series sequences for spikes, LFPs, and behavior. We used 9 sessions for pretraining the MS-Spike and MS-LFP models and held out the remaining 3 sessions as unseen data for fine-tuning and evaluation.

**Gallego-Carracedo et al., [73]**    Four monkeys (Monkeys Ch, L, H, and M) performed an instructed delay 2D center-out reaching task using a manipulandum in this dataset. Each trial began with the monkey bringing and holding the cursor at the central target. After a variable delay period, one of eight outer targets equally spaced along a circle of 6-8 cm radius was presented, followed by an auditory "go" cue that signaled the monkey to initiate a movement toward the selected target. Monkeys Ch and M were trained to wait for the auditory go cue, whereas Monkeys H and L were not. To obtain a reward, Monkeys CH and M were required to hold the cursor at the target for 0.5 seconds, whereas Monkeys H and L were required to hold for 0.1 seconds. Monkeys had to return back to the central target to start a new trial. We used 2D hand velocity in $x$ and $y$ directions as downstream behavior signals.

There were 10, 5, 3, and 5 recording sessions available for Monkeys Ch, H, L, and M, respectively, all of which were not used during the pretraining of the MS-Spike model. Unlike [63] and [64], this dataset did not include broadband signals or raw LFP signals, but instead, contained LFP power signals across 0.5-4 Hz, 4-8 Hz, 8-12 Hz, 12-25 Hz, 25-50 Hz, 50-100 Hz, 100-200 Hz, and 200-400 Hz. Therefore, we trained separate MS-LFP models on 6, 5, and 3 recording sessions of Monkeys Ch, H, and L, respectively, where we held out 4 recording sessions of Monkey Ch and all sessions of Monkey M as unseen sessions for fine-tuning. Also, this dataset included spike, LFP power, and behavior signals at 30 ms resolution, unlike the other datasets used in this study. The neural signals were recorded from M1 for Monkeys Ch and M, and area 2 of S1 for Monkeys H and L. As done for other datasets, we z-scored continuous LFP power and behavior signals for each dimension independently. All time-series data (spikes, LFPs, and behavior) were segmented into 3-second sequences before being fed into the models.

### A.3.2    Spike-only neural datasets

For the following spike-only neural datasets, we utilized the NeuroTask project, which provides a unified interface for accessing and processing the following datasets [84]. In this framework, both spike and behavior signals are provided in a trial structure with variable lengths. Although trial event annotations (e.g., go cue, target onset) were available for some, we did not perform trial alignment based on these markers and instead used the data as provided for consistency and to mimic more naturalistic settings. Trials longer than 5 seconds were segmented into sequences with a maximum duration of 5 seconds. For all datasets, we binned the spike counts within 10 ms nonoverlapping windows and used behavior signals in the same 10 ms resolution. None of the following datasets were included in the distillation analyses as they only had spike signals available, however, we performed behavior decoding analyses from the latent representations extracted by our unsupervised MS-Spike model and NDT2 (see Fig. 8 in Appendix A.6).

**Perich et al., [65–69]**    This dataset included recordings from four monkeys (Monkeys Cp, T, Mp, and J) as they performed either a 2D center-out reaching task or a 2D random target reaching task. Specifically, Monkey Cp had 51 sessions for the center-out task and 15 for the random target task; Monkey T had 6 sessions for each task; Monkey Mp had 22 center-out and 6 random target sessions; and Monkey J had 3 sessions for the center-out task and none for the random target task (109 spike-only recording sessions). For both tasks, we used 2D cursor velocity in $x$ and $y$ directions as downstream behavior signals. The neural signals were recorded from M1 for Monkeys Cp, Mp, and J, and dorsal premotor cortex (PMd) for Monkeys Cp, T, and Mp.

In the 2D center-out reaching task, the monkeys began each trial by moving their hands to the center of the workspace and was randomly presented one of eight outer targets equally spaced in a circle

after a variable waiting period. To receive a liquid reward, the monkeys were required to reach the outer target within 1 second. For the 2D random target reaching task, the monkeys reached sequentially to four targets, followed by receiving a reward. About 200 ms after reaching the target, a new target appeared, and the monkeys started the reaches immediately.

**Churchland et al., [70]**  In this dataset, two monkeys (Monkeys C1 and C2) performed a maze task on a fronto-parallel screen where they made both straight reaches and reaches that curved around one or more intervening barriers. The dataset typically included 27 different conditions, where each condition provides a particular arrangement of targets and barriers. Monkey C1 and C2 had 4 and 5 recording sessions, respectively. For both monkeys, the recordings were made from M1 and PMd. We used the 2D cursor velocity in $x$ and $y$ directions as downstream behavior signals.

**Chowdhury et al., [71]**  This dataset included recordings from three monkeys (Monkeys R1, R2, and R3) who used a manipulandum to reach to targets on a screen within a 20 cm x 20 cm workspace across two different experimental tasks. For the first experimental task, the two-workspace experiment, the screen was divided into four quadrants, and two specific regions were selected: the far ipsilateral and the near contralateral quadrants. In each trial, one of these two workspaces was randomly selected, and the monkey performed a sequence of reaches to randomly positioned targets within that workspace.

For the second experimental task, the active vs. passive center-out reaching task, each trial began with the monkey holding the cursor on a central target for a randomized duration. In half of the trials, referred to as *passive trials*, a mechanical perturbation displaced the monkey's hand toward one of four outer target directions during the center-hold period. After the perturbation (bump), the monkey actively reached to complete the trial (*active trials*). Both monkeys R1 and R2 had 2 and 3 recording sessions of center-out and two-workspace tasks, respectively, whereas Monkey R3 only had 2 recording sessions for the two-workspace task. For both tasks, the neural recordings were made from area 2 of S1, and we used the 2D hand velocity in $x$ and $y$ directions as downstream behavior signals.

**Ma et al., [72]**  In this dataset, four monkeys (Monkeys X1, X2, X3, and X4), were trained to perform three different experimental tasks where the neural signals were recorded from M1. In the first experimental task, the isometric wrist task, Monkey X4 controlled the cursor on the screen by exerting forces on a small box placed around one of the hands. Flexion and extension forces moved the cursor left and right, respectively, while radial and ulnar forces moved it up and down. Each trial began with the cursor held on the center target for a random duration, after which an auditory "go" cue prompted the monkey to reach toward one of eight outer targets. A liquid reward was given if the monkey moved the cursor to the target within 2 seconds and held it there for 0.8 seconds. For this task, we used the 2D cursor velocity in the $x$ and $y$ directions as downstream behavior signals.

In the second experimental task, the key grasping task, Monkey X1 eached toward and grasped a custom device placed beneath the screen using one hand. A pair of force-sensitive resistors measured the grasping forces applied during the task. At the start of each trial, the monkey placed its hand on a touchpad and waited for a random interval before initiating a grasp. It then moved the cursor into one of three rectangular targets displayed on the screen while simultaneously increasing and maintaining grasp force. For this task, we used the applied force in the $x$ and $y$ directions as downstream behavior signals.

In the third experimental task, Monkeys X2 and X3 performed a center-out reaching task while grasping the upright handle of a manipulandum. Each trial began with the monkey moving the cursor to the center target, followed by a variable hold period, after which it reached toward one of eight outer targets spaced uniformly around a circle. A trial was successful if the monkey reached the target within 1 second and held it for 0.5 seconds to receive a liquid reward. For this task, we again used the 2D cursor velocity in the $x$ and $y$ directions as downstream behavior signals.

## A.4  Behavior decoding

To ensure a fair comparison across all models and effectively quantify the quality of inferred latent representations, we evaluated behavior decoding by training a linear regression decoder from the inferred latent representations to the behavior signals using the training trials, and assessed

performance on the held-out test trials. To extract a single latent representation per time step, we applied mean-pooling over the latent representations of all spatial patches corresponding to the same time step. For supervised models, we used a linear layer appended to the encoder backbone as the behavior decoder. We quantified the downstream decoding performance by computing the coefficient of determination ($R^2$) between true and predicted behavior signals, averaged across behavior dimensions.

## A.5    Representation retrieval and alignment

To quantify the distillation performance, we also quantified the representation retrieval performance between paired spike and LFP representations across MS-Spike, MS-LFP, and Distilled LFP models. To achieve that, we first computed sequence-level representations by mean-pooling the latent representations of neural patches within each sequence. All metrics were computed on test sequences from recording sessions with simultaneous spike and LFP (or LFP power) recordings (see Table 2; 3,013 sequences from 65 sessions).

As our representation retrieval performance metrics, we then computed **top-1 accuracy**, **top-5 accuracy**, and **mean rank** between paired spike and LFP sequence representations. Specifically, for a given LFP sequence representation, we computed the cosine similarity with all spike sequence representations and ranked them (among 3013 sequences) in descending order of similarity. **Top-1** and **top-5** accuracies measure the proportion of LFP representations for which their corresponding paired spike representation was the most similar or was ranked among the top five most similar spike representations, respectively. Similarly, **mean rank** computes the average rank of the ground-truth paired spike representations across all LFP representations. As shown in Table 1, despite our distillation objective not being a contrastive objective—i.e., despite it not aligning paired spike and LFP representations while repelling the other unpaired spike representations—the MS-Spike and Distilled LFP models achieved high top-1/top-5 accuracy and low mean rank, indicating strong alignment. In contrast, MS-Spike vs. MS-LFP and Distilled LFP vs. MS-LFP pairs yielded much lower retrieval metrics, underscoring the effectiveness of the distillation objective. Metrics for the random baseline were computed across two randomly generated matrices with same shape of latent representations.

Surprisingly, we observed that the aforementioned retrieval metrics for the MS-Spike vs. MS-LFP and the Distilled LFP vs. MS-LFP pairs were nearly at chance level, indicating that without distillation, there is poor alignment between the spike and LFP representations at the sequence-level, i.e., when looking at alignment at a sequence-by-sequence basis. This can be counterintuitive, given that we later show that information from spike models can be effectively distilled into LFP models. To better understand this finding, in addition to the sequence-level alignment measured by the retrieval metrics above, we quantified the global alignment between the representations by computing the **linear centered kernel alignment (CKA)** scores between the representations extracted by MS-Spike, MS-LFP, and Distilled LFP models [75]. Unlike retrieval-based metrics such as top-1 accuracy, top-5 accuracy, or mean rank, which operate at the sequence-level, CKA evaluates global alignment of representational geometry as follows. Given two sets of representations matrices $X \in \mathbb{R}^{n \times d_1}$ and $Y \in \mathbb{R}^{n \times d_1}$, where $n$ is the number of sequences and $d_1$ is the representation dimensionality, CKA measures the pairwise similarity between $X$ and $Y$ by computing the Hilbert-Schmidt Independence Criterion between the centered Gram matrices $K = XX^T$ and $L = YY^T$ and normalizes this similarity to ensure invariance to isotropic scaling and orthogonal transformations:

$$\text{CKA}(X, Y) = \frac{||X^T Y||_F^2}{||X^T X||_F^2 ||Y^T Y||_F^2} \tag{2}$$

Intuitively, CKA measures whether two sets of representations encode similar relational structure, where a CKA score of 1 indicates perfect alignment, while a score close to 0 indicates dissimilar representational structure (as seen in the last row of Table 1 for the random baseline). Unlike retrieval-based metrics above, we observed that MS-Spike vs. MS-LFP, and Distilled LFP vs. MS-Spike pairs achieved substantially higher CKA scores compared to the random baseline, indicating a similar global representational geometry between them, even though they lack alignment at the individual sequence-level. However, in line with the retrieval-based metrics, the CKA score between MS-Spike and the Distilled LFP model was consistently the highest, highlighting the success of the distillation objective in aligning both local and global representational structures.

## A.6 MS-Spike vs. NDT2

As discussed in Section 3.1, we adopted an unsupervised multi-subject, multi-session modeling approach similar to NDT2 [17], which tokenizes spiking activity by grouping neurons across space into neural patches and encodes spatial variability via learned space embeddings. In NDT2, these space embeddings are **shared across sessions and subjects**, while session- and subject-specific variability is handled through **session-specific and subject-specific** tokens.

In contrast, our MS-Spike model uses **session-specific space embeddings**, where each recording session has its own learned spatial embedding for neural patches. We did not include subject- or session-level tokens, relying instead on the per-session space embeddings to account for cross-session variability. This change allows our model to flexibly represent neural dynamics across diverse sessions without enforcing a shared spatial representation across subjects.

In addition to this methodological difference, our implementation of MS-Spike differs from NDT2 in several key technical aspects—some of which also contributed to improved training efficiency:

- The transformer encoder and predictor backbones of MS-Spike utilize rotary transformers [81], whereas NDT2 utilized default transformer architectures with absolute position encodings.

- All models in our framework—including MS-Spike—support **variable-length sequences during both training and inference**. This removes the need for padding all sequences in a batch to the same length, thereby reducing both computational and memory overhead. We believe that this is a critical factor in the improved scalability of our MS-Spike model.

  For example, this enabled us to train on 5-second segments using batch sizes of 64. In contrast, we encountered severe memory issues when using the original NDT2 implementation with 5-second segments, even at very small batch sizes such as 4. These issues made training infeasible: at that rate, a single model would have required nearly a month to complete training. Attempts to use shorter sequences (e.g., 2 seconds) still failed due to memory constraints, and we ultimately trained the baseline NDT2 model using only 1-second sequences.

  As a final debugging step to resolve this issue, we observed the training curves of NDT2 with a purely supervised loss (which is computationally lighter than the unsupervised MAE objective) using 5-second segments. Even in this scenario, training still failed to converge—unlike the 1-second sequence training, which converged reliably. These findings underscore both the computational limitations of the original NDT2 framework and the practical advantages of our memory-efficient, variable-length MS-Spike implementation.

We trained our NDT2 baseline with 8 subject-specific and session-specific tokens, used a patch size ($S$) of 64 for spatial patching (same as MS-Spike), used a 10-layer transformer encoder backbone as the feature extractor, and set the masking probability to 0.6, matching the configuration used in MS-Spike. As noted above, due to memory and speed limitations, we trained and evaluated NDT2 using 1-second sequences with a batch size of 128. To ensure a fair comparison, we used **the exact same train–test splits as MS-Spike**. All other hyperparameters—including learning rate schedules—were kept at their default values as specified in the original NDT2 implementation. Training was performed in a distributed setting across 4 GPUs for approximately 400 epochs, with total training time amounting to around 125 hours.

A detailed per-subject and per-session comparison of downstream behavior decoding performance between MS-Spike and NDT2 is presented in Fig. 8. We performed the behavior decoding as described in Appendix A.4, where we inferred the latent representations from unsupervised MS-Spike and NDT2 models for the pretraining recording sessions.

Among the 226 recording sessions used in the pretraining, MS-Spike outperformed NDT2 in 208 sessions, while NDT2 performed better in 18 sessions. For sessions where MS-Spike outperformed NDT2, the average improvement in decoding performance was 0.090 decoding $R^2$. In contrast, for the sessions where NDT2 outperformed MS-Spike, the performance gain averaged only 0.014 $R^2$. Overall, MS-Spike significantly outperformed NDT2 in downstream behavior decoding ($p < 3.97 \times 10^{-37}, n = 226$, one-sided Wilcoxon signed-rank test).

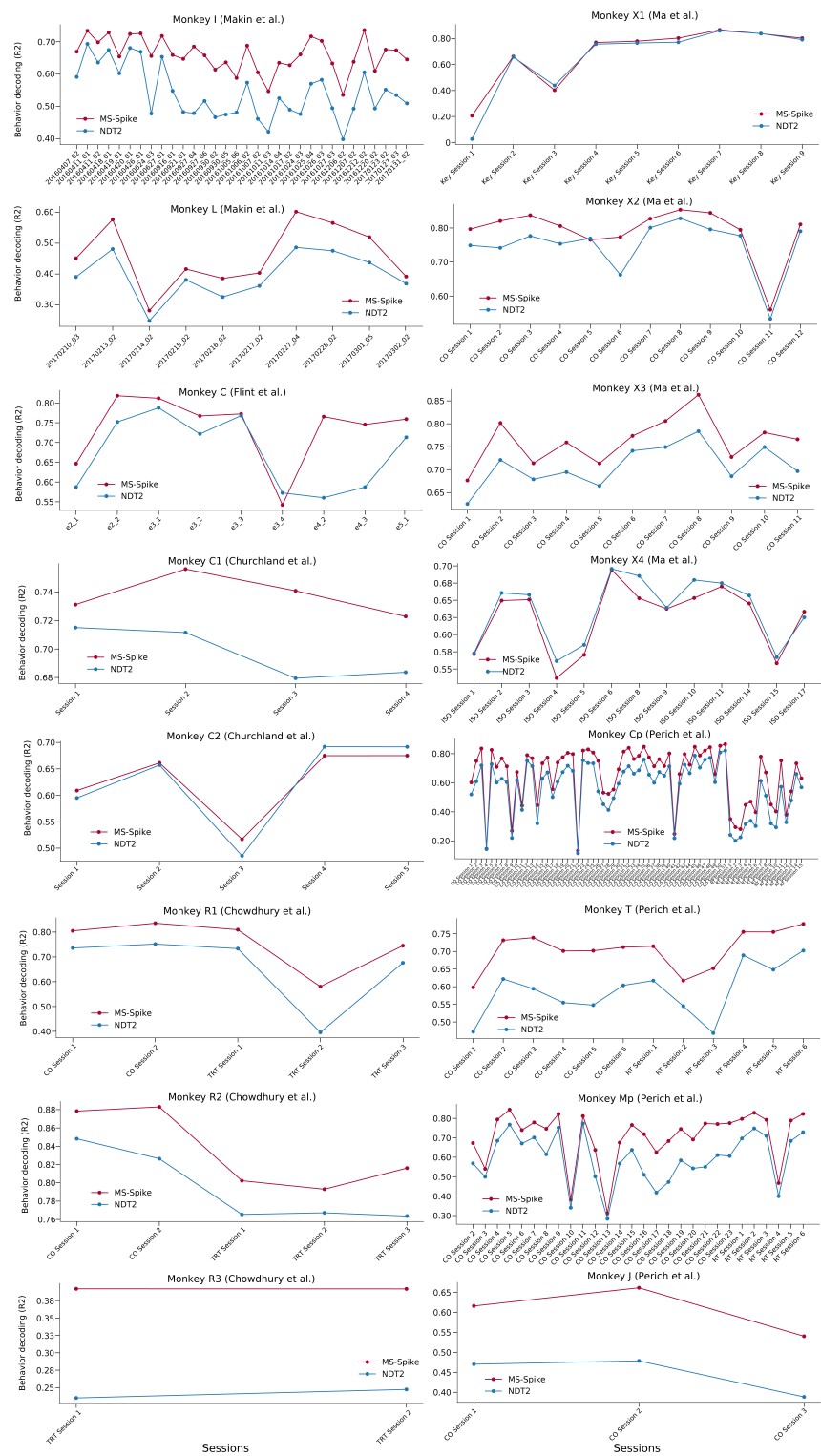

Figure 8: Per-subject and per-session downstream behavior decoding performance comparison between unsupervised MS-Spike and NDT2 models. Each subplot corresponds to a different monkey, with session-wise (denoted on the x-axis) decoding performances ($R^2$) shown. MS-Spike and NDT2 performances are directly compared for each recording session. Subject identifiers and associated dataset references are noted in the subplot titles. See Appendix A.3 for detailed dataset descriptions.

## A.7 Additional baseline comparisons with MSID and BRANT

To provide further evidence for the strong performance of our cross-modal knowledge distillation approach, we conducted two additional baseline comparisons. First, we trained multimodal models using a method termed multiscale SID (MSID) [34]—a recent multimodal framework for modeling neural activity—using spike and LFP signals from 30 recording sessions of Monkey I. Since MSID is a single-session model, a separate model was trained for each session. We used the official implementation provided by the authors and adopted the horizon hyperparameters reported in their manuscript [34] ($h_y = h_z = 10$). Under this configuration, MSID achieved an average behavior decoding performance ($R^2$) of 0.48, which is substantially lower than that of the Distilled LFP model ($R^2 = 0.66$).

Second, we attempted to fine-tune the BRANT [18] model, i.e., a multi-session model for intracranial EEG (iEEG) signals with patch-based tokenization combined with spatial electrode location embeddings, trained on the masked autoencoding (MAE) objective. Despite our extensive attempts—both freezing and updating the BRANT backbone during supervised behavior regression fine-tuning—the model did not converge using LFP signals of 30 recording sessions of Monkey I. We hypothesize that this lack of convergence may stem from 1) a substantial domain shift between the human iEEG data used for BRANT's pretraining and the nonhuman primate LFP data used in our experiments, and 2) the large scale of BRANT's architecture (~505M parameters), which likely requires substantially larger datasets for effective fine-tuning.

## A.8 Ablation studies

### A.8.1 The encoder backbone architecture

In our experiments, we used rotary transformers [81] as encoder backbones due to their flexible attention mechanism, which naturally supports pretraining objectives such as masked autoencoding involving token dropping and masked token reconstruction. In addition to this architectural advantage, we further validated our choice of using rotary transformers from a performance perspective through an ablation study comparing several encoder backbone architectures, namely long short-term memory (LSTM) networks, one-dimensional convolutional neural networks (1D CNNs), and rotary transformers. Specifically, we trained fully supervised single-session LFP models separately on 30 recording sessions from Monkey I using each of the aforementioned encoder architectures. As shown in Table 3, models with a rotary transformer backbone achieved performance comparable to those with an LSTM backbone, whereas models with a 1D CNN backbone slightly underperformed. Therefore, we adopted rotary transformers as the default encoder backbone for all subsequent experiments.

| Backbone architecture | Behavior decoding ($R^2$) |
|---|---|
| LSTM | $0.66 \pm 0.06$ |
| 1D CNN | $0.61 \pm 0.06$ |
| Rotary transformer | $0.66 \pm 0.08$ |

Table 3: Average behavior decoding performances ($R^2$) across different encoder backbone architectures for fully-supervised single-session LFP models trained on 30 recording sessions of Monkey I. $\pm$ represents standard deviation.

### A.8.2 The systematic unfreezing of the teacher MS-Spike model during cross-modal knowledge distillation

As detailed in Section 3.3, the teacher MS-Spike model was kept frozen throughout the cross-modal knowledge distillation procedure to 1) simplify training and 2) prevent representation drift in the high-performing teacher model. To validate this design choice, we conducted an ablation study on the 5 and 3 held-out recording sessions of Monkeys I and C, respectively, in which the teacher MS-Spike model was systematically unfrozen at different stages of training the Distilled LFP models: 1) from the beginning, 2) at the 10th epoch, and 3) at the 75th epoch. As shown in Table 4, the Distilled LFP models exhibited representational collapse when the teacher MS-Spike model was unfrozen and updated from the beginning (row 1), achieving an average $R^2$ of 0.18. Even though unfreezing the teacher MS-Spike model at the 10th (row 2) or 75th (row 3) epoch slightly improved

the performance, they still substantially underperformed (0.20 and 0.31 $R^2$, respectively) compared to the default configuration, where the teacher MS-Spike model was completely frozen (row 4, 0.68 $R^2$). Overall, these results confirm that keeping the teacher model frozen throughout distillation provides the most stable and effective training in our experiments, though exploring more gradual or structured unfreezing strategies may offer potential future improvements.

| Epoch unfrozen | Behavior decoding ($R^2$) |
|:---:|:---:|
| 0 | $0.18 \pm 0.08$ |
| 10 | $0.20 \pm 0.07$ |
| 75 | $0.31 \pm 0.06$ |
| - | $0.68 \pm 0.07$ |

Table 4: Average behavior decoding performances ($R^2$) of Distilled LFP models when the teacher MS-Spike model's parameters were unfrozen and updated at the epochs indicated in the *Epoch unfrozen* column. "-" denotes that the teacher model was kept completely frozen throughout training, as the default setup. $\pm$ represents standard deviation.

### A.8.3 The impact of pretraining dataset size in cross-modal knowledge distillation

A natural question arises as to whether the performance gap between the Distilled LFP and MS-LFP models is mainly driven by differences in the pretraining dataset sizes of their respective MS-Spike teacher (i.e., 226 sessions) and MS-LFP models (i.e., 34 sessions for LFP signals), rather than by inherent signal characteristics. To investigate this, we trained a new MS-Spike model (denoted as *subset*) using the same 34 recording sessions employed for MS-LFP training and used it as the teacher in the distillation process. On these 34 sessions, the MS-Spike Subset model achieved an average decoding performance of 0.54 $R^2$, outperforming the MS-LFP model (0.32 $R^2$) but underperforming the original MS-Spike model trained on all 226 sessions (0.65 $R^2$), likely due to the reduced pretraining set size. Importantly, as shown in Fig. 9, the Distilled LFP models distilled from the MS-Spike Subset teacher (denoted by Distilled LFP Subset) again outperformed the MS-LFP models on the 5 and 3 held-out sessions of Monkeys I and C, respectively, achieving $R^2$ of 0.63 for the Distilled LFP Subset model vs. 0.28 for the MS-LFP model in the 8 held-out sessions, where the teacher MS-Spike Subset achieved 0.62 $R^2$ (in the 8 held-out sessions). For reference, the original MS-Spike model trained on 226 sessions and the corresponding Distilled LFP models achieved 0.69 and 0.68 $R^2$ average decoding performance on these 8 held-out sessions, respectively. Overall, these results suggest that the performance difference between the MS-Spike and MS-LFP models is not simply due to the dataset size used during pretraining but also to the differences in signal characteristics. Critically, this gap can be overcome through cross-modal knowledge distillation.

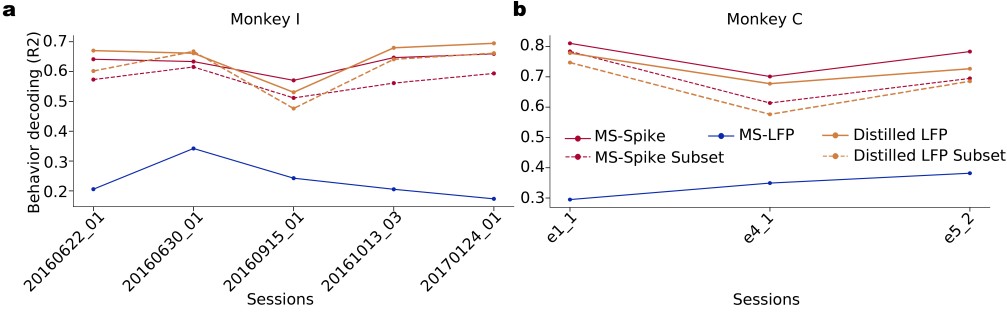

Figure 9: Behavior decoding performances ($R^2$) of the *subset* MS-Spike model and Distilled LFP models by using the fine-tuned *subset* MS-Spike models as the teachers. Performance averages are computed across all held-out fine-tuning sessions of the denoted monkeys (same sessions as in Fig. 3).

### A.9 Model-size scaling

To investigate the scalability of the cross-modal knowledge distillation approach, we trained Distilled LFP models with varying model sizes, i.e., models with 1, 2, 4, and 8 transformer encoder layers, in

addition to the default model size (transformer encoders with 10 layers). We found that the Distilled LFP models achieved superior performance to LFP-only baselines even when the model capacity was decreased 10 times.

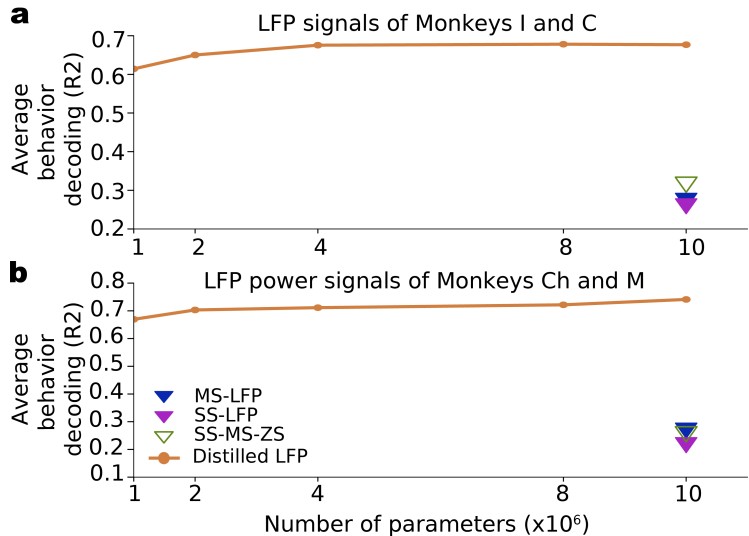

Figure 10: Average decoding performance ($R^2$) of Distilled LFP models trained on raw LFP signals from Monkeys I and C in (a) and Monkeys Ch and M in (b) at increasing parameter scales. Performance averages are computed across all held-out fine-tuning sessions of the denoted monkeys (same sessions as in Fig. 3).

In addition to evaluating the model size scalability of the Distilled LFP models, we also examined the scalability of the teacher MS-Spike model. Specifically, we trained a teacher MS-Spike model with 4 transformer encoder layers (~4M parameters) instead of 10 layers (~10M parameters), and then trained Distilled LFP models with 10 transformer encoder layers (~10M parameters) using 5 and 3 held-out recording sessions from Monkeys I and C, respectively. In this setting, the Distilled LFP models exhibited a larger performance drop compared to student model scaling, as expected, achieving an average decoding performance of 0.649 $R^2$ instead of 0.681 $R^2$.

## A.10  Cross-modal knowledge distillation performance on pretraining sessions

In addition to the fine-tuning sessions, the results of which are shown in Figs. 3 and 6, we also tested the distillation performance on the multimodal (i.e., paired spike and LFP signals) recording sessions used in the pretraining of MS-Spike and MS-LFP models for the sake of completeness. Overall, we included 25 and 9 sessions of Monkeys I and C, respectively, in the pretraining of MS-Spike and MS-LFP (trained on LFP signals) models.For Monkeys Ch, H, and La, we included 6, 5, and 3 recording sessions, respectively, in the pretraining of the MS-LFP model trained on LFP power signals.We performed both unsupervised and supervised fine-tuning of the MS-Spike model on these sessions, as these subjects were completely held out during MS-Spike pretraining (see Table 2).

**Unsupervised setting**  As done for Fig. 3, we first tested the cross-modal knowledge distillation performance in the unsupervised setting. As shown in Fig. 11, Distilled LFP models consistently outperformed all LFP-only baselines and even their teacher MS-Spike models, again suggesting that distilled models effectively leverage shared, behavior-predictive structure across spike and LFP modalities.

For example, the average behavior decoding performance ($R^2$) of the Distilled LFP model was 0.66, 0.69, 0.62, 0.77, and 0.68 for Monkeys I, C, Ch, H, and La, respectively (0.67 on average), compared to 0.63, 0.71, 0.66, 0.63, and 0.51 for the MS-Spike teacher models (0.64 on average). The MS-LFP baseline achieved considerably lower performance: 0.33, 0.34, 0.21, 0.58, and 0.37 (0.34 on average), and single-session LFP (SS-LFP) models underperformed even further, with an average of 0.28.

Consistent with results on the fine-tuning sessions, Distilled LFP models again significantly outperformed all LFP model variants ($p < 3.6 \times 10^{-15}$, $n = 48$, one-sided Wilcoxon signed-rank test), and also achieved significantly better performance than their MS-Spike teachers ($p < 5.3 \times 10^{-4}$, $n = 48$, one-sided Wilcoxon signed-rank test). Interestingly, the performance gains over the teacher MS-Spike models were particularly obvious for Monkeys H and La. This suggests that cross-modal knowledge distillation can still be beneficial even when the teacher model or teacher modality underperforms, highlighting its ability to fuse complementary information across modalities during training, despite relying solely on the LFP modality at inference time. Further, both Distilled LFP models and teacher MS-Spike models significantly outperformed the multimodal models (SS-MM and SS-MM-ZS, see Section 4 for details), highlighting the need for more complex unsupervised multimodal fusion approaches rather than input-level concatenation of multimodal signals.

**Supervised setting** Next, we compared the decoding performances in the supervised setting, following the same setup as in Fig. 6. Consistent with results obtained on fine-tuning sessions, the performance gains from cross-modal distillation remained robust, albeit smaller than those observed in the unsupervised setting as shown in Fig. 12.

For instance, the average behavior decoding performance of the Distilled LFP models was 0.75, 0.73, 0.73, 0.91, and 0.87 for Monkeys I, C, Ch, H, and La, respectively (0.77 on average), compared to 0.66, 0.71, 0.70, 0.88, and 0.82 for the MS-LFP model (0.71 on average) and 0.65, 0.69, 0.70, 0.90 and 0.87 (0.70 on average) for SS-LFP model. As expected, the distilled models again did not outperform the spike-based models (MS-Spike and SS-MM, with average performances of 0.77 and 0.79, respectively) in the supervised setting. However, the Distilled LFP models significantly outperformed all LFP-only baselines ($p < 3.8 \times 10^{-12}$ for MS-LFP, $p < 9.0 \times 10^{-13}$ for SS-LFP, and $p < 6.0 \times 10^{-13}$ for SS-MM-ZS, $n = 48$, one-sided Wilcoxon signed-rank test).

Interestingly, and consistent with the unsupervised setting, MS-Spike models fine-tuned on Monkeys H and La underperformed even the LFP-only baselines, whereas the Distilled LFP models achieved superior performance. We hypothesize that this discrepancy stems from limited generalization of the spike modality in these subjects: while MS-Spike models achieve near-perfect decoding performance on the training set, this performance does not translate to the test set, likely due to overfitting or session-specific nature in the spike signals. In contrast, the LFP-only models demonstrate stable and generalizable decoding performance across train and test splits.

Through the distillation process, the LFP models learn to better extract shared, cross-modal structure between spike and LFP representations during training, without relying on spike signals at inference time. Crucially, this process does not merely copy the MS-Spike model's outputs, but instead enables the distilled model to aggregate complementary information across modalities in a way that is more robust to train-test shifts. These results highlight the value of cross-modal distillation as a means of combining the shared, behaviorally relevant information present in both modalities with the inherent generalization stability of the LFP modality, ultimately improving downstream decoding performance.

### A.11 Cross-modal knowledge distillation performance with fully-supervised distillation

We also tested the distillation performance in a fully-supervised setting where we replaced the autoencoding objective in Eq. 1 with the behavior regression objective, resulting in the following distillation objective:

$$\mathcal{L}_{distill}^{sup} = \underbrace{\frac{1}{n_y T} \sum_{t=1}^{T} ||\boldsymbol{z}_t - f_\psi(f_\theta(\boldsymbol{y}_t))||_2^2}_{\text{Behavior regression objective}} + \lambda \underbrace{\left(1 - \frac{1}{T} \sum_{t=1}^{T} \frac{< f_\theta(\boldsymbol{y}_t), f_\gamma(\boldsymbol{s}_t) >}{||f_\theta(\boldsymbol{y}_t)||_2 ||f_\gamma(\boldsymbol{s}_t)||_2}\right)}_{\text{Representation-level alignment/similarity objective}} \quad (3)$$

where $f_\psi(\cdot)$ is a behavior regression layer used to reconstruct behavior signal $\boldsymbol{z}_t \in \mathbb{R}^2$ and $f_\gamma(\cdot)$ was fine-tuned in a fully-supervised manner (see Section A.1) to solely optimize behavior regression objective.

To provide a fairer comparison, we also fine-tuned MS-LFP models in a fully-supervised manner. As shown in Figs. 13 and 14 for recording sessions used in fine-tuning and pretraining, respectively, fully-supervised cross-modal knowledge distillation led to slight but consistent performance improvements in downstream behavior decoding performance.

For the fine-tuning sessions, fully-supervised Distilled LFP models achieved 0.81 $R^2$ average decoding performance across all subjects and sessions, significantly outperforming the Distilled LFP models in Fig. 6, which had an average performance of 0.80 $R^2$ ($p < 1.3 \times 10^{-6}, n = 51$, one-sided Wilcoxon signed-rank test). Fully-supervised fine-tuned MS-LFP models also exhibited a small gain, achieving an average of 0.76 $R^2$ compared to 0.74 $R^2$ for their supervised counterparts. However, fully-supervised Distilled LFP models still significantly outperformed fully-supervised MS-LFP models ($p < 2.6 \times 10^{-10}, n = 51$, one-sided Wilcoxon signed-rank test).

Similar trends were observed on the pretraining sessions, as shown in Fig. 14. Fully-supervised Distilled LFP models achieved an average performance of 0.774 $R^2$, significantly outperforming the Distilled LFP models from Fig. 12, which achieved 0.768 $R^2$ ($p < 4.3 \times 10^{-5}, n = 48$, one-sided Wilcoxon signed-rank test). Fully-supervised MS-LFP models also showed a slight improvement, reaching an average performance of 0.73 $R^2$ compared to 0.71 $R^2$ fo their supervised counterpart. Nevertheless, fully-supervised Distilled LFP models again significantly outperformed fully-supervised MS-LFP models ($p < 8.8 \times 10^{-11}, n = 48$, one-sided Wilcoxon signed-rank test).

Overall, these results indicate that explicit inclusion of supervised behavior regression in the distillation objective, in addition to using a fully-supervised teacher model, can further enhance the decoding performance, although the performance gains are modest compared to the improvement introduced by the distillation itself.

### A.12    Multi-session (MS) extension of cross-modal knowledge distillation

So far, we performed unsupervised pretraining of MS-Spike models and then fine-tuned them on individual recording sessions using different objectives varying in behavior supervision levels, which were later used as teacher models for the single-session cross-modal distillation setup. Next, we investigated whether this distillation process could be extended to a multi-session setting (MS-Distilled LFP) by applying the unsupervised cross-modal distillation objective across multiple sessions, thereby leveraging *large-scale unlabelled paired spike-LFP data* to pretrain a single, generalizable distilled LFP model. To achieve that, we pooled paired spike-LFP signals across multiple sessions (i.e., same sessions used for pretraining of MS-LFP model), and pretrained MS-Distilled LFP models by optimizing the **unsupervised distillation** objective in Eq. 1 where we used the unsupervised MS-Spike model as teachers (note that the pretraining of all multi-session models was also unsupervised). As done for MS-LFP models, we trained different MS-Distilled LFP models for raw LFP signals (for Makin et al., [62, 63] and Flint et al., [64] datasets), and LFP power signals (for Gallego et al., [73] dataset). For the MS-Distilled LFP model trained on LFP power signals, we first fine-tuned the teacher MS-Spike models in an unsupervised manner on spike signals of the Gallego et al., [73] dataset (note that this dataset was completely held-out during pretraining of the MS-Spike model).

After pretraining, we fine-tuned the MS-Distilled LFP models on held-out recording sessions (i.e., same held-out sessions for MS-LFP models) in three different scenarios as done for MS-Spike and MS-LFP models. First, we fine-tuned MS-Distilled LFP models in an **unsupervised manner** by optimizing the unsupervised distillation objective in Eq. 1, where we used unsupervised fine-tuned MS-Spike models as teachers. Second, we performed a **supervised** fine-tuning where we again optimized the distillation objective in Eq. 1 but used teacher MS-Spike models that are fine-tuned in a supervised manner (see Appendix A.1). Third, we performed **fully-supervised** fine-tuning by optimizing the supervised distillation objective in Eq. 3 with fully-supervised fine-tuned MS-Spike models as teachers. Compared to single-session distillation, we investigated whether unsupervised distillation-based pretraining could provide a better initialization that can further improve downstream decoding performance when fine-tuned on recording sessions with behavior labels, especially through supervised or fully-supervised fine-tuning.

As shown in Figs. 16, 17 for the recording sessions held-out for fine-tuning, MS-Distilled LFP improved performance over its single-session counterparts (Distilled LFP). For instance, in the supervised case, MS-Distilled LFP achieved 0.76, 0.78, 0.87, and 0.86 (0.82 on average) for Monkeys I, C, Ch, and M, respectively, and significantly outperformed Distilled LFP ($p < 1.3 \times 10^{-4}$, $n = 51$, one-sided Wilcoxon signed-rank test) that achieved 0.74, 0.75, 0.86, and 0.86 (0.80 on average). The performance improvements were even more significant in the fully-supervised scenario ($p < 1.5 \times 10^{-8}$, $n = 51$, one-sided Wilcoxon signed-rank test), where the MS-Distilled LFP model achieved 0.77, 0.80. 0.88, and 0.87 (0.83 on average) for Monkeys I, C, Ch, and M,

respectively, compared to 0.75, 0.78, 0.87, and 0.86 (0.81 on average). Beyond the supervised scenarios, we also compared these models in the unsupervised case and found that both models achieved comparable performances (0.71 and 0.72 on average for Distilled LFP and MS-Distilled LFP models, respectively).

It should also be noted that the recording sessions of Monkey M were completely held-out during the pretraining of the MS-Distilled LFP model, as done for the MS-LFP model. For Monkey M, we performed single-session fine-tuning of the pretrained MS-Distilled LFP model in two steps: 1) fine-tuning the MS-Spike model—-the teacher model used during pretraining of the MS-Distilled LFP-—on Monkey M's spike data, and 2) subsequently fine-tuning the pretrained MS-Distilled LFP model on Monkey M's LFPs using the fine-tuned MS-Spike model from step 1 as the teacher. Through this pipeline, we aimed to transfer the spike-LFP alignment learned across animals to a new subject. Therefore, the results for Monkey M shown in Figs. 16 and 17 (bottom right panels) also demonstrate the generalization capability of the MS-Distilled LFP model to an unseen subject through fine-tuning.

We also performed the same comparisons for the recording sessions used in pretraining of MS-Distilled LFP models, as they had behavior labels available. However, it is important to note that this may not always be the case in practical scenarios, where large-scale multimodal neural recordings are often unaccompanied by corresponding behavior signals. Similar to held-out fine-tuning sessions, we found that MS-Distilled LFP models again outperformed Distilled LFP models. In the supervised setting, MS-Distilled LFP models achieved significantly superior performance compared to Distilled LFP models ($p < 9.6 \times 10^{-8}$, $n = 48$, one-sided Wilcoxon signed-rank test) where they achieved 0.79 vs. 0.75, 0.76 vs. 0.73, 0.72 vs. 0.73, 0.91 vs. 0.91, and 0.88 vs. 0.87 for Monkeys I, C, Ch, H, and La, respectively (0.79 vs. 0.77 on average). Similarly, the performance improvements were again more significant in the fully-supervised setting ($p < 1.3 \times 10^{-12}$, $n = 48$, one-sided Wilcoxon signed-rank test) where MS-Distilled LFP and Distilled LFP achieved 0.79 vs. 0.76, 0.77 vs. 0.74, 0.76 vs. 0.71, 0.92 vs. 0.92, and 0.90 vs. 0.90 for Monkeys I, C, Ch, H, and La, respectively (0.80 vs. 0.77 on average). Unlike fine-tuning sessions, MS-Distilled LFP significantly outperformed Distilled LFP also in the unsupervised setting ($p < 6.1 \times 10^{-8}$, $n = 48$, one-sided Wilcoxon signed-rank test), where they achieved 0.69 vs. 0.66, 0.72 vs. 0.69, 0.63 vs. 0.62, 0.78 vs. 0.77, and 0.67 vs. 0.68 for Monkeys I, C, Ch, H, and La, respectively (0.70 vs. 0.67 on average).

Overall, these results demonstrate that multi-session cross-modal distillation offers a powerful extension to the single-session distillation by leveraging shared structure across recordings to produce stronger and more generalizable LFP models. While single-session distillation already yields substantial improvements over LFP-only baselines, incorporating broader variability through multi-session distillation-based pretraining further enhances the distilled model's capacity to capture modality-shared, behaviorally predictive features. This advantage is particularly pronounced in supervised and fully-supervised fine-tuning regimes, where multi-session distilled models consistently outperform their single-session counterparts.

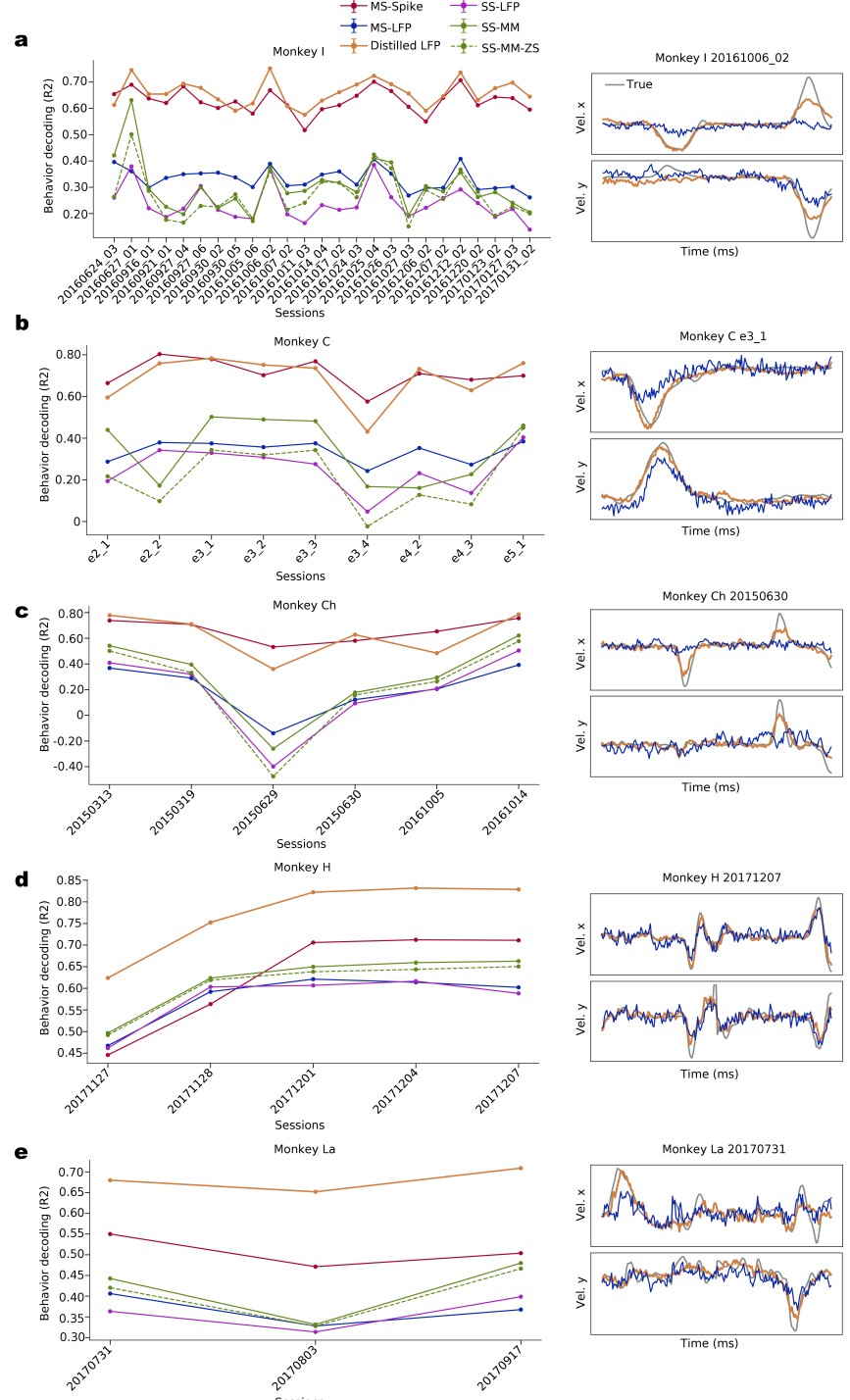

Figure 11: Behavior decoding performances ($R^2$) of unsupervised models on the recording sessions that were used in **pretraining** of MS-Spike and MS-LFP models. Each panel (a-e) shows decoding results across sessions for individual monkeys (Monkeys I, C with LFP signals, and Monkeys Ch, H, and La with LFP power signals) and example decoded behavior trajectories from latent representations of Distilled LFP models and MS-LFP models on these sessions. The MS-Spike model was fine-tuned for Monkeys Ch, H, and La, as these subjects were completely held out during MS-Spike pretraining. In contrast, no fine-tuning was applied for the plotted sessions here from Monkeys I and C, as these monkeys' other sessions were used during MS-Spike pretraining. For the MS-LFP models, no fine-tuning was performed for any session, since all sessions with LFP signals were included during the MS-LFP models' pretraining.

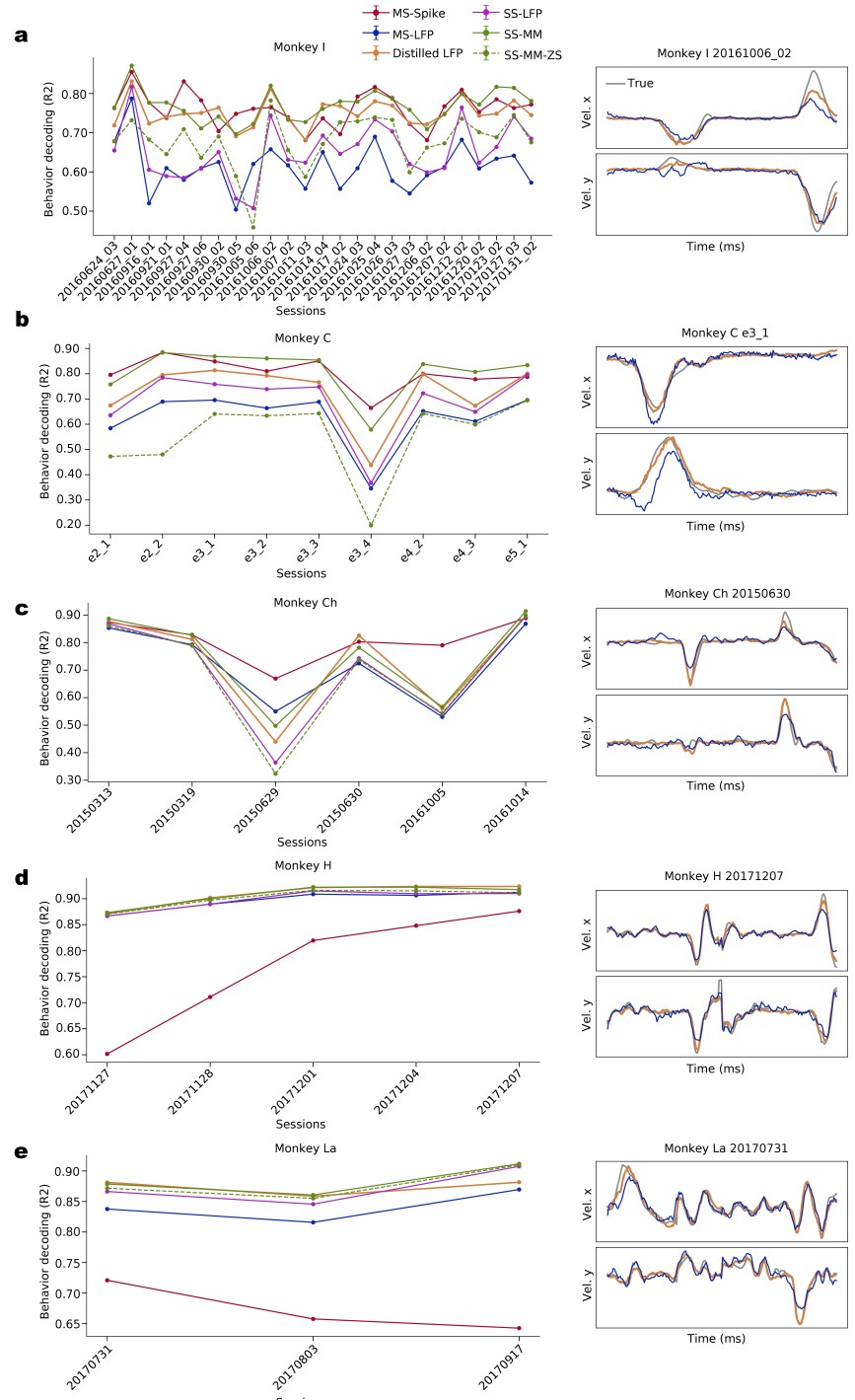

Figure 12: Behavior decoding performances ($R^2$) of supervised models on the recording sessions that were used in **pretraining** of MS-Spike and MS-LFP models. Figure conventions are the same as in Fig. 11. Unlike Fig. 11, however, MS-Spike and MS-LFP models shown here were fine-tuned in a supervised manner (note these underwent unsupervised pretraining).

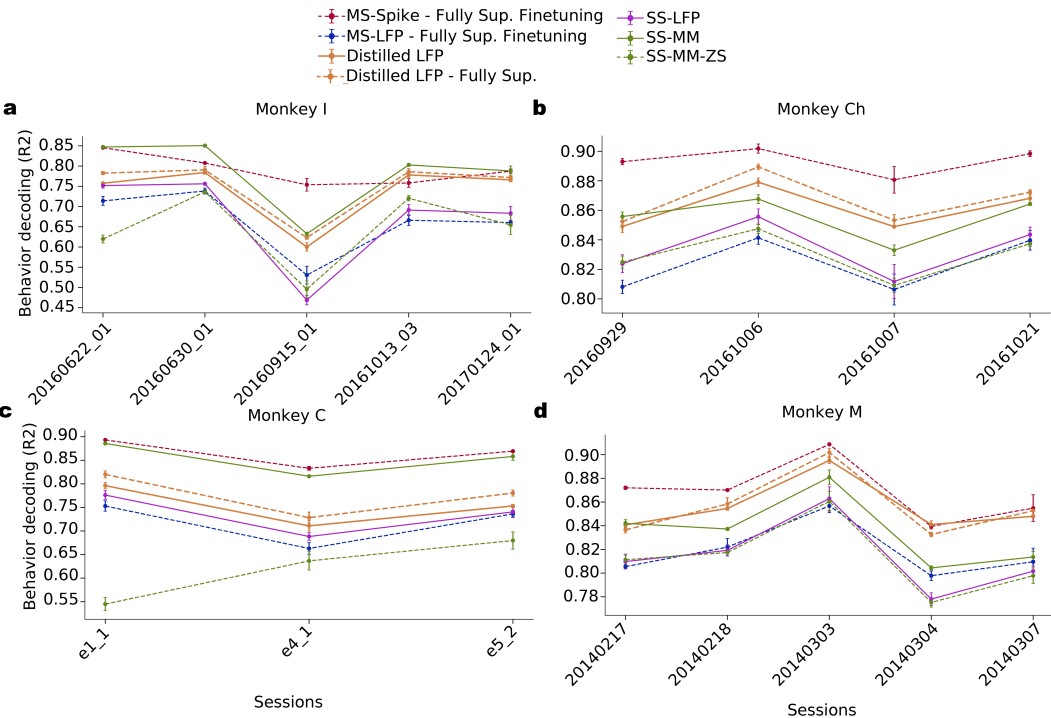

Figure 13: Behavior decoding performances ($R^2$) in fully-supervised (Fully Sup.) setting. Each panel (a-d) shows the decoding results across sessions for individual monkeys (Monkeys I, C with LFP signals, and Monkeys Ch, H, and La with LFP power signals). Unlike Fig. 6, we demonstrate MS-Spike and MS-LFP models that are fine-tuned in a fully-supervised manner just with the behavior regression objective, and Distilled LFP models that are trained via the supervised distillation objective in Eq. 3 with fully-supervised MS-Spike models being teachers. Single-session models are trained only through the behavior regression objective, as done for Figs. 6, 12.

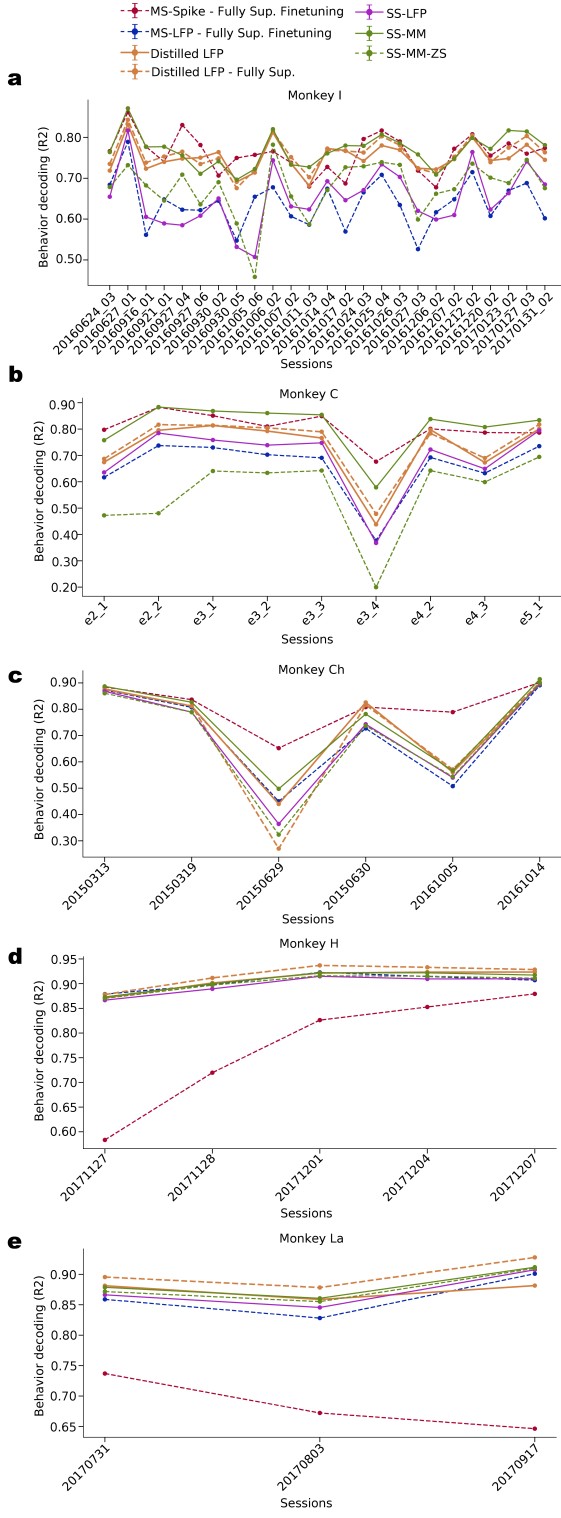

Figure 14: Behavior decoding performances ($R^2$) in fully-supervised (Fully Sup.) setting on the recording sessions that were used in **pretraining** of MS-Spike and MS-LFP models. Each panel (a-e) shows decoding results across sessions for individual monkeys (Monkeys I, C with LFP signals, and Monkeys Ch, H, and La with LFP power signals). Unlike Fig. 12, we demonstrate MS-Spike and MS-LFP models that are fine-tuned in a fully-supervised manner just with the behavior regression objective, and Distilled LFP models that are trained via the supervised distillation objective in Eq. 3 with fully-supervised MS-Spike models being teachers. Single-session models are trained only through the behavior regression objective, as done for Figs. 6, 12, 13.

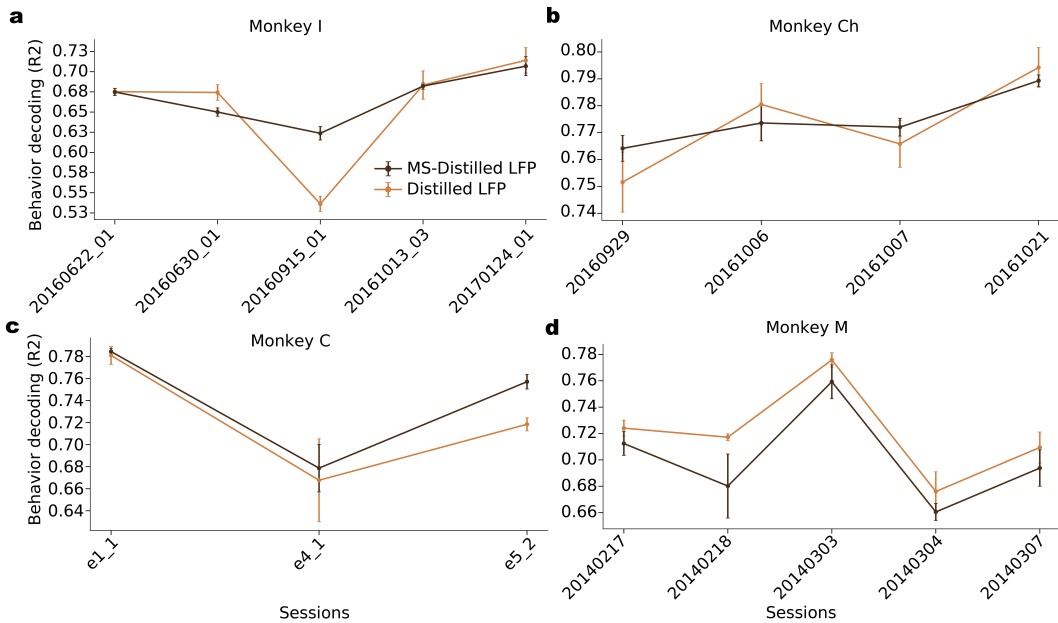

Figure 15: Behavior decoding performances ($R^2$) of unsupervised MS-Distilled LFP and Distilled LFP models. Both models were trained (and then fine-tuned for the MS-Distilled LFP model) through the unsupervised distillation objective in Eq. 1, where teacher MS-Spike models were fine-tuned also in an **unsupervised** manner.

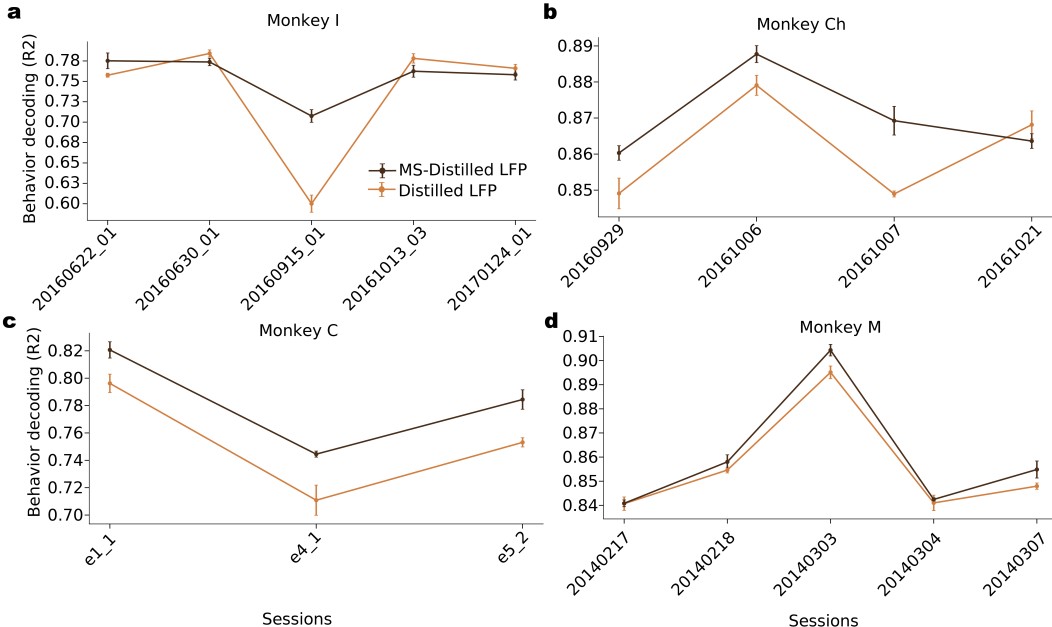

Figure 16: Behavior decoding performances ($R^2$) of supervised MS-Distilled LFP and Distilled LFP models. Both models were trained (and then fine-tuned for the MS-Distilled LFP model) through the unsupervised distillation objective in Eq. 1, while teacher MS-Spike models were fine-tuned in a **supervised** manner.

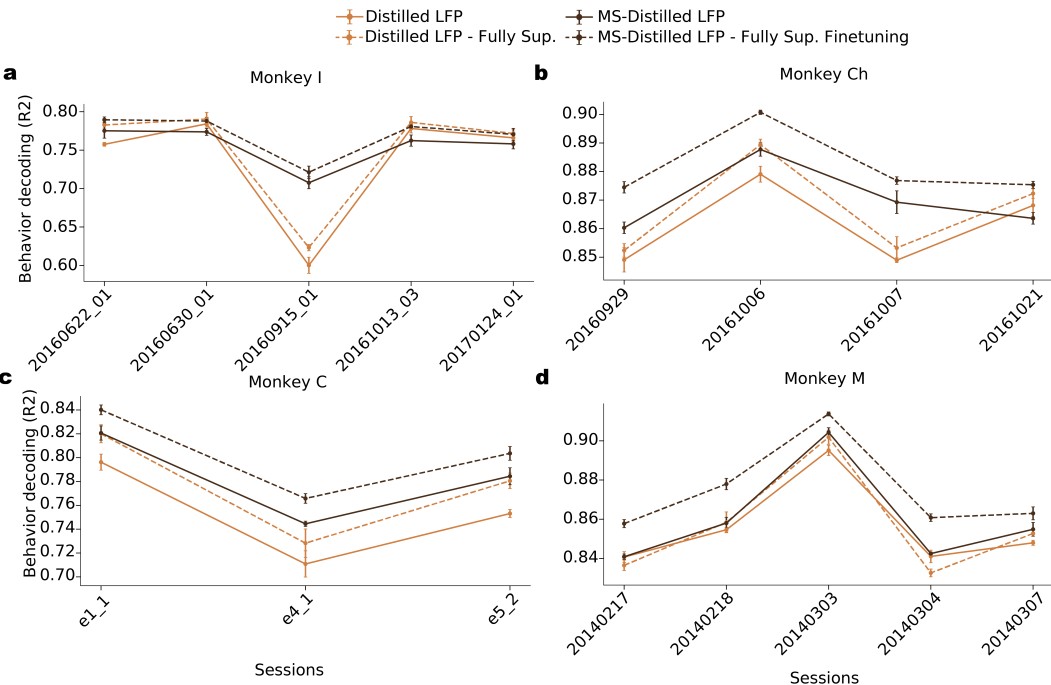

Figure 17: Behavior decoding performances ($R^2$) of fully-supervised MS-Distilled LFP and Distilled LFP models. Both models were trained (and then fine-tuned for the MS-Distilled LFP model) through the **supervised** distillation objective in Eq. 3, and the teacher MS-Spike models were fine-tuned also in a **fully-supervised** manner.

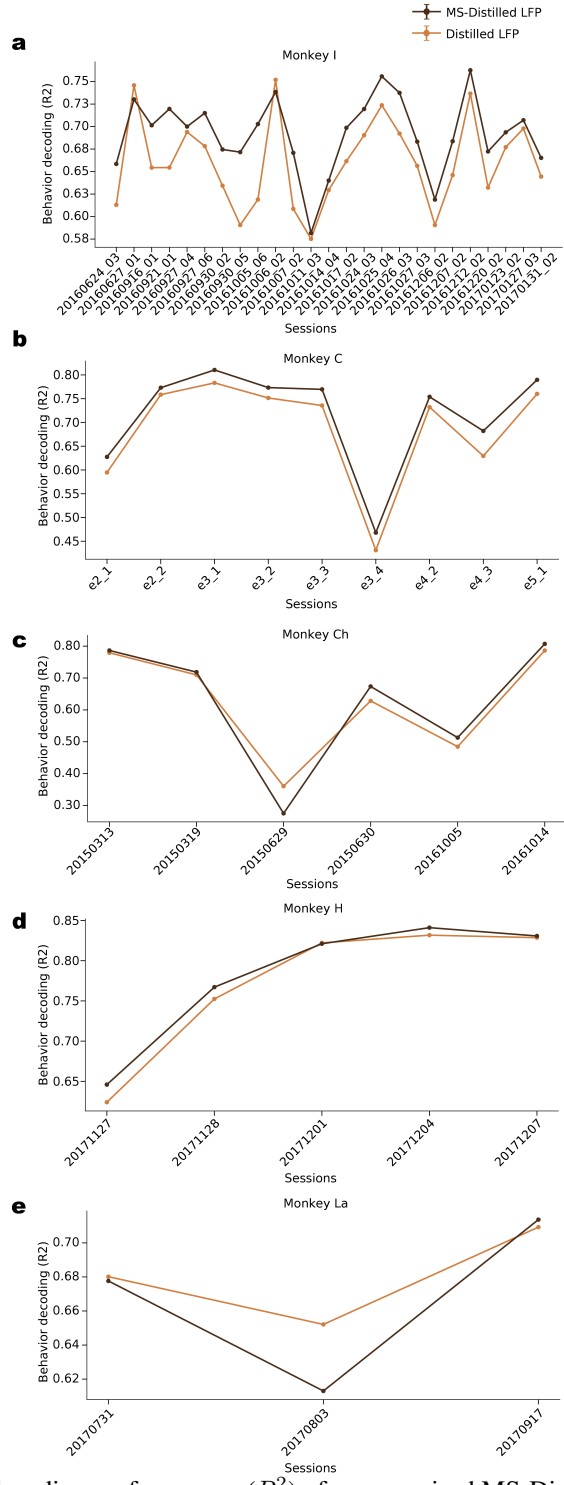

Figure 18: Behavior decoding performances ($R^2$) of unsupervised MS-Distilled LFP and Distilled LFP models on the recording sessions that were used in **pretraining** of MS-Spike, MS-LFP, and MS-Distilled LFP models. Figure conventions are the same as in Fig. 15.

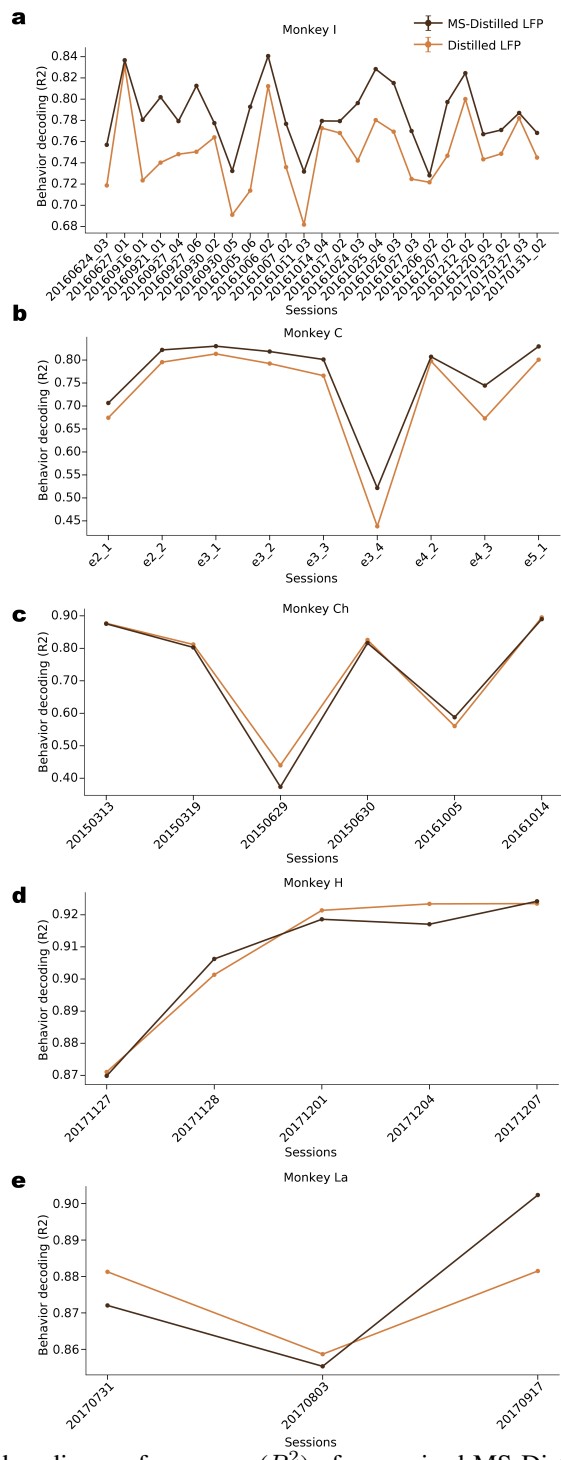

Figure 19: Behavior decoding performances ($R^2$) of supervised MS-Distilled LFP and Distilled LFP models on the recording sessions that were used in **pretraining** of MS-Spike, MS-LFP, and MS-Distilled LFP models. Figure conventions are the same as in Fig. 16.

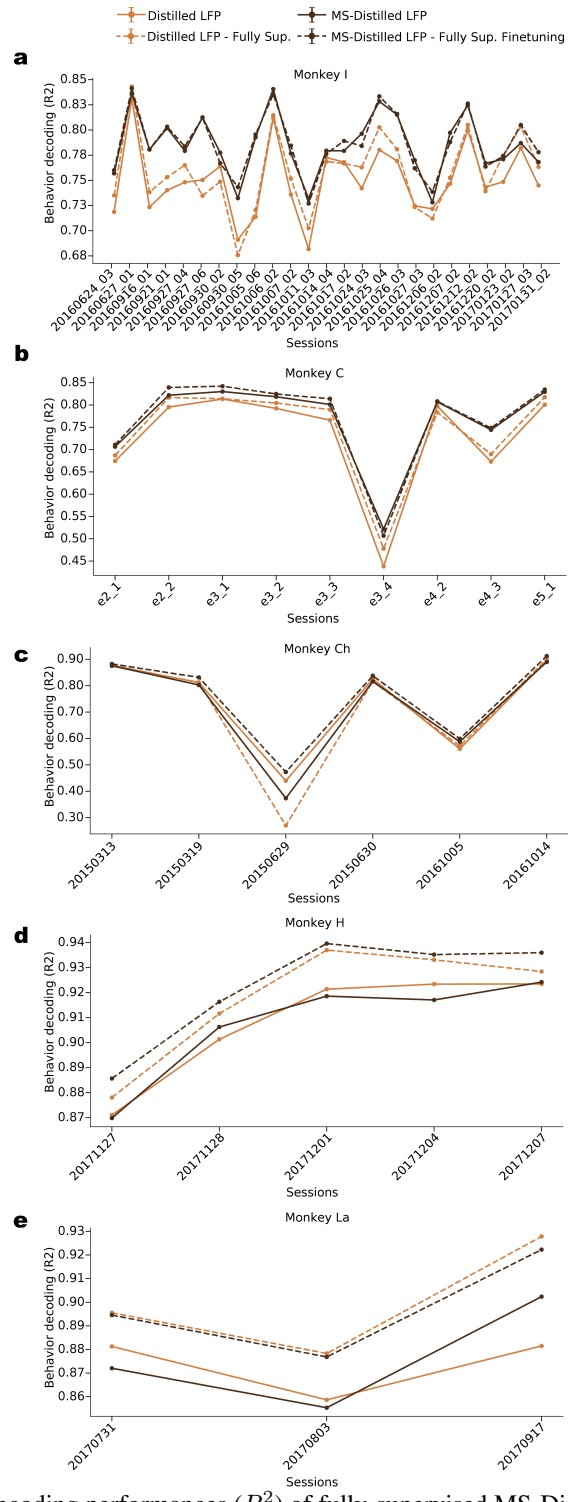

Figure 20: Behavior decoding performances ($R^2$) of fully-supervised MS-Distilled LFP and Distilled LFP models on the recording sessions that were used in **pretraining** of MS-Spike, MS-LFP, and MS-Distilled LFP models. Figure conventions are the same as in Fig. 17.

