# OpenReview forum: "Cross-Modal Representational Knowledge Distillation for Enhanced Spike-informed LFP Modeling"
_NeurIPS.cc/2025/Conference — NeurIPS 2025 poster_

### Official Review · Reviewer_ZtJZ · 2025-06-27

**Clarity:** 3
**Significance:** 2
**Originality:** 3
**Rating:** 5
**Confidence:** 2

**Summary:**

The authors propose a cross-modality distillation framework that leverages existing neural-spike based transformer decoders to boost the performance of LFP-based transformer decoder. The authors argue that having LFP-based transformer decoders is also important, as LFP recordings can be more easily obtained in some cases. They essentially train a LFP-based transformer by combining autoencoding loss (to ensure the latent representations can reconstruct the LFP signal) and a cosine loss (to align the latent representations of the LFP transformer decoder to the neural spike transformer decoder). The authors compared their distillation-based LFP transformer with reasonable baselines (e.g., vanilla LFP-based transformer) across multiple datasets. They show significant improvement in decoding performance of the distilled LFP-transformer in comparison to the baselines. Finally, they extend their distilled LFP-based transformer to a multi-session approach demonstrating modest improvements in decoding (increase of 0.01 and 0.02 in R2 coefficient, although statistically significant).

**Questions:**

Out of my own curiosity, have the authors tried not keeping the weights of the teacher model not fixed in the distillation objective? It might be possible that allowing the LFP signal to influence the representations of the spike-based transformer might lead to a better representation.

**Ethical Concerns:**

["NO or VERY MINOR ethics concerns only"]

**Final Justification:**

Authors response has not impacted my score. My opinion of the work remains the same.

**Limitations:**

Yes

**Quality:**

3

**Strengths And Weaknesses:**

**Strengths**
1. The paper is well-written and does a good job of explaining its problem. It is important to also develop techniques that can work with LFP data which is more readily available than its neural spike counterpart.

2. The paper proposes an extension of NDT2 that performs significantly better than NDT2 (increase in behavior decoding (R2 coefficient) by roughly 0.9). The proposed extension changes the tokenization of the spiking data to allow the transformer to leverage spatial relationship in addition to the temporal ones.

3. The paper uses a cross-modal knowledge distillation framework that uses neural spike-based transformer (trained with larger datasets) to train single session LFP transformer. The corresponding distilled LFP transformer significantly outperforms transformers trained using only LFP data robustly showing the value of using distillation.

4. The authors perform extensive validation of their technique against reasonable baselines, and show consistent improvement.

**Weakness**

1. The improvements of distilled LFP models over the multi-session spike-based transformer are modest (improvement of 0.01 to 0.02 in the R2 coefficient). Since, the latent representation of the distilled LFP model is trained to be similar to the spike-neural transformer, it is not surprising that the distilled LFP model and spike-based transformers have similar decoding performance as the decoding is performed using latent representations. The fact that adding LFP data does not yield a substantial improvement seems to suggest that most of the decoding capability is resulting from the spike-based representations with the LFP data having a minor effect on improving the representation.

2. The multi-modal baseline used for comparison is a bit naive. Concatenating the LFP signal and the neural spike is one of the simplest way to perform multi-modal learning. There are a lot more principled ways to perform multimodal learning, e.g., take a look at the review by Xu et al. IEEE Trans. Pattern Recognition'23. I am not expecting authors to compare their method with new multi-modal approaches, as it is not the main point of the work. I would just encourage authors to acknowledge the limitation of their multi-modal approach and discuss that the another approach to improving LFP models would be through multi-modal learning (there are multi-model learning approaches that also allow unimodal predictions). Indeed, with enough availability of both LFP data and spike-based data, it could be possible that multi-modal approaches could outperform the distillation approaches.

---

> ### Author Rebuttal · Authors · 2025-07-30
>
> We thank the reviewer for their thoughtful and constructive evaluation of our work. We appreciate their recognition of our contributions and address their comments below.
>
> > Substantial improvement over MS-Spike models
>
> As the reviewer noted, distilled LFP models do not yield substantial gains in behavior decoding compared to their teacher MS-Spike counterparts. This observation is expected and aligns with prior findings that spike signals alone outperform LFP signals in decoding, which is precisely why we chose the spike model as the teacher for distillation. However, we emphasize that our goal is to enable a high-performance unimodal LFP model, which does not need to use any spiking activity for decoding. As such, the value of our approach lies not in surpassing spike-based decoding but in enabling improved decoding just from LFP signals—**a more practical, more robust, and less invasive modality—while narrowing the gap to spike-based performance**. Further, please see item 3 in our response to reviewer KUPu, where we show that the distilled LFP models can still achieve a competitive decoding performance on a new session without any further finetuning or distillation.
>
> > Multimodal modeling
>
> We thank the reviewer for this thoughtful comment. We agree that our multi-modal baseline—simple concatenation of spike and LFP signals at the input level—is indeed a naive approach compared to more principled multi-modal learning strategies. We acknowledge this limitation and note that more sophisticated multi-modal approaches could be explored as a complementary direction, particularly when both spike and LFP data are available at inference time (L967–968).
>
> However, we would also like to highlight an important challenge in this setting. Multi-modal models can degrade significantly when the higher-quality modality (spikes) is missing at inference, sometimes performing worse than unimodal baselines (Ma et al., 2022). This issue becomes even more pronounced in unsupervised multi-modal modeling, which is crucial for training generalizable models, where the lower-quality modality can dominate training, making it harder to learn robust and generalizable representations. In contrast, our cross-modal distillation framework is explicitly designed to enable high-quality inference from LFP signals alone, without requiring spikes at test time. This is critical given the practical advantages of LFPs and that spike recordings degrade over time. Please see the important applications of an LFP-alone model in the introduction (L30-38).
>
> Taking these considerations together, we believe our cross-modal distillation offers **a simpler, more scalable, and more robust approach** for the specific problem of improving LFP decoding when spike data is unavailable at inference, which is a critical scenario in neuroscience and neurotechnology applications. This argument is further supported by our new analyses.
>
> > Unfreezing the teacher network
>
> We thank the reviewer for their question! Per the reviewer’s request, we performed an experiment on unfreezing the teacher network at different stages of the distillation training. When the teacher was **updated from the start of distillation**, the model suffered from **representational collapse**, achieving only $0.09~R^2$ on session 20160622_01 of Monkey I. We then tested a **delayed unfreezing schedule** by unfreezing the teacher network at 10th or 75th epochs of the distillation. While these prevented collapse, they resulted in slightly inferior performance: unfreezing the teacher network at 10th and 75th epochs achieved $0.668 \pm 0.072$ and $0.674 \pm 0.066$ average $R^2$ over 8 sessions (5 finetuning sessions of Monkey I and 3 finetuning sessions of Monkey C as in Fig. 3), where freezing the teacher network completely (**default setting**) achieved $0.681\pm0.067$ $R^2$.
>
> We would also like to point the reviewer to prior work showing that freezing the teacher network during distillation leads to improved student performance when the teacher is fine-tuned (Abbaspourazad et al., 2025; Li et al., 2025). Despite this prior evidence and our experience, we acknowledge that an alternative scheduling algorithm outperforming the completely frozen scenario can be developed, which we will leave as an interesting future direction.
>
> References:
> - Ma et al. (2022). Are Multimodal Transformers Robust to Missing Modality? IEEE/CVF Conference on Computer Vision and Pattern Recognition (CVPR)
> - Abbaspourazad et al. (2025). Wearable Accelerometer Foundation Models for Health via Knowledge Distillation. arXiv.
> - Li et al. (2025). Distilling Knowledge from Large-Scale Image Models for Object Detection. ECCV.

---

> > ### Comment · Reviewer_ZtJZ · 2025-08-04
> >
> > I thank the authors for their response. A couple of points.
> >
> > 1. I would not call LFP a less-invasive modality. One would typically require an ECoG array implanted in the brain to record either spikes or LFP, hence, I would call them equally invasive. Indeed, the same electrodes are recording the LFP signals and spikes. Now, I buy the argument that LFP recordings might be more stable than spike recordings, but like any neural recordings, LFP recordings would also degrade. Physically, the instability in the spike data comes from electrode degradation, change of underlying neural dynamics, which would also affect the LFP data. Now, the main advantage of LFP signal emphasized in the work is its robustness, but none of the experimental work actually test that the degraded LFP signal are still able to decode while the degraded spike data does not. Hence, I would tone down the applicability of LFP signals as a competent replacement of the spike-based transformers, as that aspect remains to be tested.
> >
> > 2. A fundamental limitation of the cross-modal distillation approach (from the results) is that it cannot be significantly better than its teacher model, whereas with multi-modal approaches (despite the challenges, which are not particular to LFP and Spike-based data and have been addressed in different fields) there exists possibility to improve beyond what each modality can achieve. Hence, I still see multi-modal approaches as a promising alternative. I would refrain from calling the cross-modality distillation approach as a simpler, more scalable, and more robust approach, without adequate experimental evidence, which is not present in this work.

---

> > > ### Author Response · Authors · 2025-08-04
> > >
> > > We thank the reviewer for raising several great points here.
> > >
> > > > Robustness of LFPs
> > >
> > > First, we agree that implants recording LFPs are indeed still invasive in the sense that they require a surgery/craniotomy. By less invasive in the response, we simply meant more durable over time/stable (see next point), which we agree is a much better word here. We thank the reviewer for noting this point. Indeed, we currently don’t (and won’t) use the word invasive anywhere in the manuscript.
> > >
> > > Second, we are glad the reviewer agrees that LFP recordings might be more stable than spike recordings, which is a primary motivation for developing unimodal LFP models, as also done in prior work (Flint et al., 2013; Milekovic et al., 2018). Indeed, while LFP signals also degrade over time as noted by the reviewer, prior work has shown that this degradation is significantly slower than spikes and thus LFPs are more stable over time (Flint et al., 2016; Wang et al., 2014). This higher stability has been attributed to factors such as electrode tip encapsulation that hinders discerning action potentials (in 300-5000 Hz), whereas LFPs, which are lower-frequency signals (<250 Hz), can still be recorded (Heldman and Moran, 2020) or to LFPs being less sensitive to the loss or drift of individual neurons as LFPs reflect population-level synaptic activity (Sharma et al., 2015). Also interestingly, prior work has shown successful decoding from LFPs even after spikes have degraded (Wang et al., 2014). Nevertheless, as the reviewer notes, the publicly available datasets do not provide the recordings after spike degradation. As such, we agree that the applicability of LFP models as a competent replacement of the spike-based transformers after spike degradation remains to be tested in future work/datasets.  We thank the reviewer for raising this great point and will note it under discussion. What this manuscript shows is that in the available datasets, the proposed cross-modal knowledge distillation approach enables a significant improvement in unimodal LFP decoding.
> > >
> > > > Multimodal modeling
> > >
> > > We agree with the reviewer that multimodal models are a promising alternative when both modalities are available/usable at inference time, and can, in principle, outperform unimodal LFP models by leveraging complementary information in LFP and spike signals. We will state this in our discussion. However, as observed previously and referenced above, in many applications such as BCIs, access to high-fidelity spike signals may degrade over time, or spikes may become unstable and thus hard to utilize. Thus, we also wanted to highlight the practical challenge of building models that remain effective when the high-fidelity modality (spikes) is unavailable or degraded. As noted in prior work (e.g., Ma et al., 2022), such scenarios can lead to significant performance drops in multimodal models if not carefully addressed (see also our comparisons between SS-MM and SS-MM-ZS that motivate future sophisticated multimodal approaches). By robust, we were referring to the robustness of LFPs to electrode degradation compared with spikes, as referenced above. By simple, we meant that the cross-modal approach does not need complex multimodal fusion architectures but instead follows a simple similarity-level alignment approach. By scalable, we meant the approach only requires a single modality with lightweight processing at inference time. That being said, as stated above, we acknowledge that replacing spike-based models completely with competent LFP-based models requires more future work. We will discuss these points and the potential of multimodal approaches as alternatives in the discussion. Again, we sincerely thank the reviewer for raising these points.
> > >
> > > References:
> > >
> > > -	Flint et al. (2016). Long-Term Stability of Motor Cortical Activity: Implications for Brain Machine Interfaces and Optimal Feedback Control. JNE.
> > > -	Flint et al. (2013). Long term, stable brain machine interface performance using local field potentials and multiunit spikes.  JNE.
> > > -	Milekovic et al. (2018). Stable long-term BCI-enabled communication in ALS and locked-in syndrome using LFP signals. Journal of Neurophysiology.
> > > -	Sharma et al. (2015). Time Stability and Coherence Analysis of Multiunit, Single-Unit and Local Field Potential Neuronal Signals in Chronically Implanted Brain Electrodes. Bioelectronic Medicine.
> > > -	Wang et al. (2014). Long-term decoding stability of local field potentials from silicon arrays in primate motor cortex during a 2D center out task. JNE.

---

### Official Review · Reviewer_KUPu · 2025-07-03

**Clarity:** 2
**Significance:** 3
**Originality:** 3
**Rating:** 4
**Confidence:** 4

**Summary:**

The authors propose a method for learning behavioral-predictive representations of LFPs by first pretraining a spiking model and then distilling the learned representations into an LFP model. Additionally, session-specific tokens are used to align latent representations across recording sessions. In experiments predicting behavior, the distilled LFP models perform roughly as well as spike models, and outperform single- and multi-session LFP-only baseline models. A TSNE embedding and an analysis with centered kernel alignment confirm that the distilled LFP model achieves similar representations to the multi-session spike model. The main result also holds up in a supervised finetuning setting and a multi-session distillation setting.

**Questions:**

Are my assessments in "Strengths and Weaknesses" about the smaller LFP dataset and the necessity of paired LFP-spike data accurate?

Can you say anything specific about the features of LFPs that allow the distilled LFP models to predict behavior about as well as the spike models?

Minor comments:
* The experiments section should state the number of tokens for pretraining and finetuning,

**Ethical Concerns:**

["NO or VERY MINOR ethics concerns only"]

**Final Justification:**

My two major reservations with original paper have been addressed. First, I find the LFP-only results (MS-LFP) much easier to interpret with the addition of the MS-Spike subset results, and the distilled LFP subset results should also help with the interpretation. Second, the new generalizability results to a completely unseen session demonstrate a stronger form of generalizability than the original paper. For these reasons, I have raised my score.

**Limitations:**

yes

**Quality:**

3

**Strengths And Weaknesses:**

The idea of learning informative LFP features by distilling knowledge from a spike model is very attractive given all the benefits of LFPs listed in the abstract (stability, lower power requirements, robustness) and the generally higher predictive performance of spikes in various behavior prediction scenarios.

The transformer architecture trained using session-specific learnable tokens and a masked autoregressive objective is a reasonable and flexible approach.

The finding that an LFP-only model can predict behavior as well as a spiking model by aligning its representations is very interesting to a computational neuroscience audience.

The MS-LFP data has many fewer sessions than the MS-Spike data (34 vs. 226). It seems plausible that this could explain the poor performance of the LFP-only models. Additionally, I would expect the LFP data that is only available as LFP power to be less informative than the LFPs themselves. I find it hard to interpret the results of the LFP-only baselines given these considerations.

In the first paragraph of the Results section, there is a discussion about testing generalization to unseen subjects. But "after pretraining, we fine-tuned the MS-spike model on the spikes of the held-out sessions," so it seems that there are no truly unseen subjects at the time of testing the distilled LFP model. Additionally, the results in section 4.3 are not on true hold-out sessions or animals, since they are used in pretraining. If I am following correctly, it seems the strategy taken to test the proposed method is not applicable to the case where only LFPs are available, with no shared spikes. But this is the exact scenario where the proposed knowledge distillation would be expected to be most beneficial and practical.

Given these concerns in the previous two paragraphs, I find it difficult to come away with a clean interpretation of the results of the paper.

---

> ### Author Rebuttal · Authors · 2025-07-30
>
> We thank the reviewer for their acknowledgement of our work’s strengths and its importance for the computational neuroscience community. Below, we answer the reviewer’s questions:
>
> > Number of different sessions across MS-LFP and MS-Spike
>
> We appreciate this insightful observation and agree that the smaller number of sessions in the MS-LFP dataset (34 vs. 226 for MS-Spike) can indeed be a contributing factor to the performance gap between the LFP-only and spike-only models.
> However, this imbalance stems not from our experimental design but from **the limitations of publicly available datasets—a key motivation for our cross-modal knowledge distillation from spikes to LFPs**. As noted in L26–28, most motor neural datasets have focused on spike recordings, likely due to theirr higher signal-to-noise ratio (SNR) as well as more memory-efficient data sharing. In contrast, LFP signals are typically derived from broadband recordings sampled at high frequencies (e.g., 30 kHz), which require much larger storage (often 1–8 GB per session in datasets like Makin et al.), whereas spike timing data for the same session can be shared in a much smaller storage (100 MB–1 GB). Given this data availability difference in the neuroscience domain, recent modeling approaches largely utilized spike signals to train models on larger datasets and provide more rigorous evaluations.
>
> Given these constraints, **we did our best to include as many publicly available datasets containing both spike and LFP signals as possible**. While the current availability of LFP data is limited, we believe that with larger-scale LFP datasets and more sophisticated pretraining strategies, it would be possible to develop more accurate MS-LFP models in the future. However, even in a future large-scale setting, we believe that the proposed cross-modal distillation would offer advantages due to the SNRs of spikes and LFP signals. As supporting evidence, we trained an MS-Spike model using the same 34 sessions as MS-LFP, which achieved $0.544\pm0.083$ $R^2$, outperforming MS-LFP ($0.325\pm0.042$ $R^2$) but underperforming the MS-Spike model trained on 226 sessions ($0.650\pm0.065$ $R^2$), likely due to the reduced training set. This aligns with prior work showing spikes often carry more behaviorally relevant information, as reflected in higher decoding performance (Bansal et al., 2011). Therefore, using spike models to guide LFP models can help denoise the LFP signal and improve its behavioral relevance.
>
> > LFP vs. LFP power features
>
> We understand and agree with the intuition that raw LFP signals should, in principle, contain at least as much information as the LFP power features, since the latter are computed from the former. As expanded below, the reason for using LFP power in some datasets was that in those datasets, raw LFP signals were not available.
>
> We would like to highlight that the **LFP baselines trained on raw LFP signals and those trained on LFP power features are not directly comparable**, as they originate from different datasets collected during different experiments. Specifically, recordings from monkeys Ch, H, La, and M were obtained from the dataset by Gallego-Carracedo et al., where only preprocessed LFP power features are publicly available. Consequently, separate models had to be trained on these sessions. Thus, these results also reflect performance differences across distinct datasets and preprocessing pipelines rather than a strict comparison of LFP signals versus LFP power features.
>
> > Generalizability
>
> The distillation is performed on a single-session basis by using the LFP signals of that session, after fine-tuning the pretrained MS-Spike model on the spike signals of that session, which is then used as the teacher network during distillation. Thus, it is correct that during the test time of the distilled LFP models, there are no ‘truly’ unseen subjects or sessions; when we say ‘unseen’, **we refer to the being ‘unseen’ in the pretraining of MS-Spike models**. We would like to highlight that **this is the case for other multi-session and multi-task models of neural activity, such as NDT2 or POYO**, i.e., before testing the pretrained model on a new subject/session, some degree of fine-tuning is required for all of these models as they contain subject/session-specific parameters that need to be learned. **In this manner, the reviewer’s understanding is correct, and our work does not differentiate from the prior work in this regard, but follows a similar pretraining/fine-tuning strategy.**
>
> Reviewer’s understanding of results in Section 4.3 is also correct. The spike signals of sessions on the x-axis of Fig. 5 are indeed used during pretraining of the teacher MS-Spike model. However, **during the distillation objective, we use the LFP signals of a single session (20160622_01 of Monkey I)**, after fine-tuning the pretrained MS-Spike model on that same session’s spike signals to use it as a teacher network. Thus, the LFP signals of the sessions on the x-axis of Fig. 5 are never seen during parameter updates/training; **they are just used for inference**. In Fig. 5, we show that the distillation objective learns a spike-LFP alignment that generalizes beyond the recording session on which the distillation is performed.
>
> Nevertheless, we understand the reviewer’s concern, and we sincerely thank them for bringing it up. **Now, we pretrained a new MS-Spike model where we first held out the recording sessions shown in the tables below (in addition to the sessions on which we performed distillation)**. Then, we performed the following:
>
> 1. Pretrain a new MS-Spike model where all sessions shown in tables below are held out, in addition to session 20160622_01 of Monkey I, and all sessions of Monkey C.
> 2. Fine-tune the MS-Spike model on spike signals of session 20160622_01 of Monkey I and session e1_1 of Monkey C separately
> 3. Use fine-tuned MS-Spike models from step 2 as teacher and perform distillation with LFP signals of session 20160622_01 of Monkey I and session e1_1 of Monkey C separately (so distillation is only performed on single sessions)
> 4. Perform inference on the LFP signals of sessions below (held-out).
>
> In this setting, we obtained the following results. Note that **‘Distilled LFP’ and ‘MS-LFP’ refer to results in Fig. 5**, whereas ‘New Distilled LFP’ refers to the new model trained as described above.
>
> Monkey I:
> |Session|MS-LFP|Distilled LFP on 20160622_01|New Distilled LFP on 20160622_01|
> |-|-|-|-|
> |20160624_03|0.396|0.607|0.598|
> |20160627_01|0.361|0.579|0.539|
> |20160916_01|0.298|0.576|0.529|
> |20160921_01|0.336|0.566|0.423|
> |20160927_04|0.350|0.562|0.482|
> |20160927_06|0.353|0.599|0.558|
> |20160930_02|0.355|0.570|0.582|
> |20160930_05|0.338|0.617|0.562|
> |20161005_06|0.301|0.450|0.452|
> |20161006_02|0.389|0.571|0.541|
> |20161007_02|0.306|0.563|0.547|
> |20161011_03|0.310|0.546|0.504|
> |20161014_04|0.348|0.600|0.567|
> |20161017_02|0.360|0.537|0.546|
> |20161024_03|0.310|0.611|0.582|
> |20161025_04|0.406|0.661|0.640|
> |20161026_03|0.352|0.655|0.587|
> |20161027_03|0.267|0.510|0.445|
> |**Mean**|**0.341**|**0.577**|**0.538**|
>
> Monkey C:
> |Session|MS-LFP|Distilled LFP on e1_1|New Distilled LFP on e1_1|
> |-|-|-|-|
> |e2_1|0.287|0.499|0.495|
> |e2_2|0.379|0.628|0.639|
> |e3_1|0.375|0.696|0.691|
> |e3_2|0.357|0.629|0.664|
> |e3_3|0.375|0.648|0.653|
> |e3_4|0.242|0.356|0.388|
> |e4_1|0.340|0.528|0.556|
> |e4_2|0.342|0.598|0.599|
> |e4_3|0.273|0.494|0.497|
> |e5_1|0.384|0.625|0.673|
> |e5_2|0.384|0.585|0.560|
> |**Mean**|**0.340**|**0.572**|**0.583**|
>
> In this setting, the generalization performance of the new distilled LFP of Monkey I model slightly decreased compared to the one in Fig. 5; but, it still significantly outperformed the MS-LFP model. Unlike Monkey I, the distilled LFP model of Monkey C outperformed its counterpart in Fig. 5 and MS-LFP. These results indicate that the our distillation approach can generalize to recording sessions **even when they are completely unseen, i.e., neither of the spike signals nor the LFP signals are used during training or even fine-tuning**. This result also shows that even for new sessions that do not contain paired spikes, the distilled LFP models can still offer an advantage.
>
> We believe that this result is a great addition to our manuscript, and we sincerely thank the reviewer for their great insight on recommending that.
>
> > About LFP features that allow superior performance for LFP signals
>
> Prior work in neuroscience shows that LFP and spike signal dynamics have common dynamical modes that are dominantly predictive of downstream behavior (Abbaspourazad et al., 2021). Given this evidence, aligning the LFP representation to spike representation through the distillation objective likely allows the learning of the behaviorally relevant dynamical features in LFP time-series data.
>
> > Number of tokens
>
> The number of tokens processed during pretraining and finetuning of MS-Spike is 120.1M and 4.8M, respectively. We will include these details in our manuscript.
>
> Overall, we thank the reviewer for their careful evaluation and for highlighting important points, which we addressed in detail with new analyses and clarifications. We hope that these responses and the new evidence provided will help the reviewer to reassess their overall evaluation of our work.
>
> References:
> - Stavisky et al. (2015). A high performing brain–machine interface driven by low-frequency local field potentials alone and together with spikes. JNE.
> - Abbaspourazad, et al. (2021). Multiscale low-dimensional motor cortical state dynamics predict naturalistic reach-and-grasp behavior. Nature Comm.
> - Ahmadipour et al. (2024). Multimodal subspace identification for modeling discrete-continuous spiking and field potential population activity. JNE.
> - Bansal et al. (2011). Relationships among low-frequency local ﬁeld potentials, spiking activity,
> and three-dimensional reach and grasp kinematics in primary motor
> and ventral premotor cortices. Journal of Neurophysiology.

---

> > ### Comment · Reviewer_KUPu · 2025-08-04
> >
> > I thank the authors for their detailed and constructive response.
> >
> > **Interpreting LFP-only results**
> > I recognize that data availability is a major obstacle to answering many of these questions. However, these additional numbers (0.65 $R^2$ MS-Spike, 0.54 MS-Spike subset, 0.33 MS-LFP) clearly demonstrate the performance gap between the spike and LFP models (~0.21 $R^2$). Then the poor performance of the LFP-only models **is not** just due to the differences in data quantity.
> >
> > However, I think it would be informative to train an additional "Distilled LFP subset" model that only uses this subset of sessions. This should provide an answer **what portion of the performance gap between spike and LFP models can be overcome by distillation** in a fairer apples-to-apples comparison. Given the results presented in Figure 3, I would expect most of the performance gap to be overcome.
> >
> > **Generalizability**
> >
> > > ... for new sessions that do not contain paired spikes, the distilled LFP models can still offer an advantage.
> >
> > I think this additional experiment substantially improves the paper, providing evidence for improved performance when a session is recorded without spikes. This is a stronger form of generalization than the results in the original paper.
> >
> > Thank you for providing the additional context that NDT2 and POYO also require session- and subject-specific parameters.
> >
> > **Overall thoughts**
> > My two major reservations with original paper have been mostly addressed. First, I find the LFP-only results (MS-LFP) much easier to interpret with the addition of the MS-Spike subset results, and the distilled LFP subset results should also help with the interpretation. Second, the new generalizability results to a completely unseen session demonstrate a stronger form of generalizability than the original paper. For these reasons, I have raised my score.
> >
> >
> > Minor comments that don't need any response:
> > * LFPs are known to better generalize across subjects. As a direction of future work, it would be interesting to explore whether session and subject tokens could be removed or made uninformative from the LFP encodings while still achieving good predictive performance.
> > * Where I was going with the question about which specific LFP features improve performance: It would be interesting to see if an unsupervised LFP-only preprocessing model could achieve some of the performance gains seen in this paper, perhaps by learning about which features of LFPs are distilled.

---

> > > ### Author Response · Authors · 2025-08-05
> > >
> > > We sincerely thank the reviewer for their careful investigation of our new results and for raising their score.
> > >
> > > > An additional "Distilled LFP subset" model that only uses this subset of sessions
> > >
> > > This is another great suggestion from the reviewer. Per their comment, we now ran the distillation procedure by using the **MS-Spike Subset from our rebuttal as the teacher model** and performed the same fine-tuning and distillation procedure for the sessions in the tables below. In this scenario, we obtained the following results:
> > >
> > > |Subject-Session|MS-LFP in Fig. 3|MS-Spike Subset|Distilled LFP Subset|
> > > |-|-|-|-|
> > > |Monkey I - 20160622_01|0.223|0.573|0.602|
> > > |Monkey I - 20160630_01|0.349|0.615|0.667|
> > > |Monkey I - 20160915_01|0.264|0.514|0.480|
> > > |Monkey I - 20161013_03|0.208|0.554|0.629|
> > > |Monkey I - 20170124_01|0.177|0.594|0.662|
> > > |Monkey C - e1_1|0.294|0.783|0.746|
> > > |Monkey C - e4_1|0.344|0.614|0.576|
> > > |Monkey C - e5_2|0.374|0.694|0.685|
> > > |**Mean**|**0.279**|**0.618**|**0.631**|
> > >
> > > whereas we had the following result in Fig. 3:
> > >
> > > |Subject-Session|MS-Spike in Fig.3|Distilled LFP in Fig.3|
> > > |-|-|-|
> > > |Monkey I - 20160622_01|0.636|0.675|
> > > |Monkey I - 20160630_01|0.643|0.674|
> > > |Monkey I - 20160915_01|0.585|0.536|
> > > |Monkey I - 20161013_03|0.652|0.683|
> > > |Monkey I - 20170124_01|0.676|0.714|
> > > |Monkey C - e1_1|0.814|0.781|
> > > |Monkey C - e4_1|0.701|0.668|
> > > |Monkey C - e5_2|0.773|0.718|
> > > |**Mean**|**0.685**|**0.681**|
> > >
> > > Consistent with the reviewer’s expectation, the performance gap between MS-Spike Subset and MS-LFP is indeed overcome through the distillation. In this scenario, the performance of the distilled LFP model (Distilled LFP Subset) is slightly reduced when using the MS-Spike Subset as the teacher, compared to the results shown in Fig. 3. This drop is likely due to the smaller dataset used to train the teacher model. Nevertheless, the distilled LFP models still outperform the MS-LFP baseline by a substantial margin. As the reviewer noted, these results offer a more direct, apples-to-apples comparison and further demonstrate the effectiveness of our distillation approach beyond differences in dataset size.
> > >
> > > Once again, we sincerely thank the reviewer for their careful and thoughtful feedback during this rebuttal process and for raising their score. Their suggestions have strengthened our manuscript, and we agree that the proposed future directions are valuable for understanding generalization and the learning dynamics of the distillation procedure. We will incorporate these into our discussion as important future directions.

---

> > > > ### Comment · Reviewer_KUPu · 2025-08-06
> > > >
> > > > I applaud the authors for providing this additional experiment, which provides strong evidence that roughly all of the performance gap between spikes and LFPs can be overcome by distillation, and that this is not due to the differences in the quantity of data for the two modalities.

---

### Official Review · Reviewer_LV5f · 2025-07-04

**Clarity:** 3
**Significance:** 3
**Originality:** 3
**Rating:** 4
**Confidence:** 4

**Summary:**

The paper introduces an unsupervised cross-modality distillation framework for electrophysiology, combining both spikes and local field potential (LFP). They first train a teacher spike model across multiple recording sessions using a masked autoencoding objective with a session-specific neural tokenization strategy, and then align the latent representations of the student LFP model to those of the teacher spike model. When tested on motor cortical data from monkeys, their distilled LFP model consistently outperforms the other LFP baselines. In some of the sessions tested, they find some small improvements in performance of the spike only model, suggesting that including both modalities can enhance the performance of BCI. They provide evidence that the model can align modalities while also capturing individual differences in the latent embeddings.

**Questions:**

1. How do other existing LFP baselines compare? Please provide comparisons with other traditional deep learning methods (e.g., MLP, CNN, LSTM/state space model), as well as iEEG or other LFP-based transformer baselines to assess how performant a rotary-transformer is for LFP modalities.
2. How does your distillation approach compare to a multi-modal training approach? Please compare with multimodal approaches like those described in Schanechi and Pesaran’s work on the topic.
3. Can you please clarify the size of patches for the LFP vs. the spikes? How is the data split into train and test? Please describe the model evaluation procedure in detail for both modalities.
4. How general is your distillation framework? Could it be applied to another decoding model (e.g. MtM, POYO)?
5. What is the performance degradation while downsizing the teacher model instead of the student model?

**Ethical Concerns:**

["NO or VERY MINOR ethics concerns only"]

**Final Justification:**

During the response period, the authors provided new experiments that demonstrate the utility of their method in across day and animal transfer, and attempted to provide new unimodal baselines to contextualize their results. These new experiments gave further confidence in the approach and that the results are meaningful.

**Limitations:**

Yes

**Quality:**

3

**Strengths And Weaknesses:**

**Strengths:**
- The work addresses an important challenge and aims to utilize LFP which is often discarded from analysis, and can be used with minimal preprocessing steps
- Novel approach to use cross-modality distillation for improving decoding from LFPs
- The results suggest that pretraining with spikes helps LFP decoding, and the incorporation of LFP can also help unimodal BCI decoding in some cases.

**Weaknesses:**
- While the results in Figure 3 demonstrate a clear advantage of the distilled multimodal model over the LFP only decoder, it is not clear whether this is a strong enough LFP baseline for this dataset. Additional baselines are needed to demonstrate the advantage of the multimodal approach. There are existing cross-modal decoders from the Schanechi and Pesaran groups, and a number of LFP-based transformer models in the literature (Brant, NeurIPS 2023; Seegnificant, NeurIPS 2024) that could be used to build a more compelling set of baselines.
- Clarity of methods.  The entire pretraining method and finetuning is described in 3.1. This should be broken up into different subsections that delineate the pretraining objective from your downstream evaluations. Section 3.2 is also not very easy to follow. Instead of connecting back to the MAE pretraining in 3.1, the reader is pointed to pretraining in Appendix. It would be useful to give a more clear step by step guide to your method and what remains frozen vs. learned, and the dynamics of updates across the two models.
- It is unclear whether the model has learned generalization information about the relationship between modalities, or uses LFP as additional animal-specific features. Demonstration of some across-modality transfer to new animals would be helpful to understand whether the relationship between modalities can generalize to new settings.
- The advantage of distillation over multi-modal pretraining has not been discussed. How critical is it to pretrain with spikes and then start learning with LFP in smaller model vs. pretraining with the objective in end-to-end manner? Why have two separate models
There should be some alignment across modalities but also LFP carries different information, do you assume this is not useful for the downstream task and is discarded? Do you have evidence that this alignment objective is useful for decoding? Does it help to denoise the LFP?
- Often, student teacher models require specific learning schedules. How do you unfreeze or add in supervision over training?
- Patching over neurons doesn’t really make sense as the ordering of neurons has no specific meaning. Please justify this choice of tokenization.

---

> ### Author Rebuttal · Authors · 2025-07-30
>
> > Additional baselines and expanding the distillation approach to other models
>
> We thank the reviewer for thoughtful suggestions. As requested, we incorporated new baselines for further validations.
>
> First, we trained MSID (Ahmadipour et al., 2024)—a multimodal model of neural activity from suggested groups—on 30 sessions from Monkey I. MSID achieved a mean $R^2$ of $0.48\pm0.09$, **substantially lower than our unsupervised distilled LFP models**, which achieved $0.66\pm0.05$. The underperformance of MSID is consistent with its reliance on linear latent dynamics.
>
> Second, we attempted to fine-tune the BRANT model, using the publicly available checkpoint. Despite extensive attempts—both freezing and updating the BRANT backbone—the model failed to converge on the Monkey I dataset. We hypothesize this is due to:
> - **Domain shift** between the human SEEG data used for pretraining and the NHP LFP data used in our setting, and
> - **Model scale**—BRANT’s ~505M parameters likely require larger datasets to fine-tune effectively.
>
> Additionally, for LFP data without distillation, we compared our rotary transformer backbone, 10-layer LSTMs, and 10-layer 1D CNNs (all producing 256-dimensional embeddings), trained in the supervised setting. Across 30 sessions of Monkey I:
> |Model|Decoding $R^2$|
> |-|-|
> |LSTM|$0.66\pm0.06$|
> |1D CNN|$0.61\pm0.06$|
> |Rotary Transformer|$0.66\pm0.08$|
>
> While these results are comparable, our supervised distilled LFP models still achieved significantly higher performance ($0.76\pm 0.04~R^2$) than all of them. We selected the transformer backbone primarily for its compatibility with masked autoencoding objectives and its strong performance across diverse deep learning tasks.
>
> While performance is important, we emphasize that the **core contribution of our work** lies in its **conceptual advance**—enabling **high-performance unimodal LFP decoding through cross-modal distillation**, rather than simply aiming to outperform existing LFP models by changing the model architecture.
>
> Importantly, our framework is **architecture-agnostic**: the distillation approach could be applied to architectures such as BRANT, MtM, NDT2, or POYO in the future.
>
> > Clarity of methods, adding supervision, freezing
>
> We thank the reviewer for helpful suggestions on the write-up, and we apologize for the confusion caused. In Section 3.2, we originally referred to Appendix A.1 for MS-Spike fine-tuning details. However, we will now also include a clear summary of these details directly in Section 3.1, where they are most relevant. We will also add a step-by-step guideline for training/finetuning details, summarized as follows:
>
> 1. Pretraining an MS-Spike model on pretraining sessions
> 2. Finetuning the MS-Spike model on one of the finetuning tasks, depending on the level of supervision. **For supervised fine-tunings, we add a regression head to the pretrained MS-Spike model, as the pretraining was unsupervised. During all fine-tunings, all model parameters are updated.**
> 3. Initializing an LFP model, then training it via the cross-modal knowledge distillation objective where the teacher model is the fine-tuned MS-Spike model from step 2. **During this step, the teacher MS-Spike model is fully frozen (as described in L208-209).**
>
> > Learning schedules
>
> We chose to keep the teacher MS-Spike model fully frozen during distillation. The decision was based on simplicity and prior evidence (Abbaspourazad et al., 2025; Li et al., 2025).
>
> As described in response to reviewer ZtJZ item 3, we tested the impact of unfreezing the teacher model. When the teacher was **updated from the start of distillation**, the model suffered from representational collapse, achieving only $0.09~R^2$ on one session of Monkey I. We then tested a **delayed unfreezing schedule**, where the teacher network remained frozen for the 10 or 75 epochs and was updated thereafter. While these prevented collapse, they achieved slightly worse performance.
>
> > Multi-modal modeling
>
> As highlighted in L100–101, our goal is distinct from multi-modal modeling: rather than leveraging both LFP and spike signals at inference time, we aim to build **high-performing LFP-only models** (the important applications for LFP-only models are provided in the introduction L30-38).
>
> As also discussed in L24–28, our motivation to use a pretrained spike-based teacher is twofold:
>
> - Spike models (e.g., NDT2, MS-Spike) are well-established as SOTA for motor decoding (L39–41),
> - Public spike datasets are significantly more abundant than multimodal datasets, enabling more generalizable pretraining. (L27-28).
>
> Another advantage of cross-modal knowledge distillation over multi-modal learning is its simplicity, i.e., not requiring complex alignment approaches as in many multi-modal works (Peng et al., 2025; Wang et al., 2019). This is particularly important when one modality is of lower quality than another (i.e., LFP vs. spikes) (Huang et al., 2021).
>
> We agree that comparing against multi-modal baselines is important, and thus, we provided comparisons against 2 multimodal baselines: SS-MM and SS-MM-ZS (see L231-237). As shown in our results, distilled LFP models consistently outperform the SS-MM-ZS baseline (Figs. 3,4,6,10,11,12,13). Further, we now show that our distillation approach outperforms a multi-modal baseline, MSID.
>
> > Denoising & discarding information of LFP
>
> The reviewer is correct that the distillation objective contributes to denoising the LFP signals. However, we **do not believe this process completely discards LFP-specific information** that may be complementary. We highlight 3 reasons supporting this:
>
> - **We include an autoencoding objective with the distillation loss**, encouraging the model to retain intrinsic LFP information.
> - **The distillation loss does not converge to zero** (e.g., 0.8 cosine similarity), suggesting that the model retains some degree of LFP-specific encoding.
> - As shown in Fig. 3, distilled LFP models can even **outperform their teacher MS-Spike models** on certain sessions. This would be unlikely if important LFP-specific information were discarded.
>
> > Generalization
>
> We do demonstrate generalization to a new subject in Section 4.5, where we evaluate **multi-session (MS) distillation approach** on a **held-out animal (Monkey M)**. The procedure is as follows:
>
> 1. Pretrain an MS-Spike model by including some sessions of Monkeys Ch, H, and La (in addition to the 226 sessions), **excluding any data from Monkey M**.
> 2. Using the MS-Spike model from step 1 as the teacher, pretrain an MS distilled LFP-power model through the distillation objective by excluding Monkey M
> 3. Fine-tune the MS-Spike model from step 1 on Monkey M’s spikes
> 4. Fine-tune the MS distilled LFP model from step 2 on Monkey M’s LFPs, where the fine-tuned MS-Spike model from step 3 is used as the teacher.
>
> This pipeline **transfers the spike-LFP alignment learned across animals** to a new subject, and **the resulting distilled LFP models generalize well to Monkey M**. We will make this clearer in the manuscript.
>
> We would also like to ask the reviewer to kindly see item 3 in response to reviewer KUPu, showing generalizability to fully unseen recording sessions with no training/finetuning, and distillation.
>
> > Patching, data details, evaluation
>
> We agree that the ordering of spiking neurons does not inherently carry semantic meaning. While the neuron index is arbitrary, the patched groups still represent neurons recorded from the same cortical region (M1) with shared and complementary dynamics. The learnable patch embeddings and the self‑attention mechanism allow the model to flexibly capture these correlations without assuming any fixed spatial structure. A similar patching mechanism can also be found at NDT-2.
>
> We provided patch sizes for spike and LFP signals across different settings in Appendix A.2 (L645-646, L649-651), where we used a patch size of 64 for spikes, and a size of 32 and 288 for LFP and LFP power signals, respectively. These choices are made for computation efficiency, as small patch sizes increase the input sequence lengths.
>
> For data splits, we perform a random 80/10/10% train/validation/test split. Models are evaluated based on downstream behavior decoding: we train a linear regression head on the embeddings produced by the encoder, detailed in Appendix A.4. During evaluation, we combine validation & test sets to improve the robustness of performance statistics, particularly for sessions with limited data.
>
> > Teacher model size
>
> Per reviewer’s request, we now performed distillation from a 4M parameter MS-Spike model, where the student distilled LFP models had 10M parameters. Mean performance in Fig. 9a, where both teacher MS-Spike and student distilled LFP models had 10M parameters, was $0.681$ $R^2$. With a 4M parameter MS-Spike model, we obtained a slightly worse mean performance of $0.649~R^2$, and a larger drop compared to student model scaling as expected, but distilled LFP models again outperformed other LFP-only baselines.
>
> Overall, we hope the further evidence above on our model’s performance and generalization answers the reviewer’s concerns and encourages the reviewer to reconsider our work.
>
> References:
> - Ahmadipour et al. (2024). Multimodal subspace identification for modeling discrete-continuous spiking and field potential population activity. Journal of Neural Engineering.
> - Abbaspourazad et al. (2025). Wearable Accelerometer Foundation Models for Health via Knowledge Distillation. arXiv.
> - Li et al. (2025). Distilling Knowledge from Large-Scale Image Models for Object Detection. ECCV.
> - Peng et al. (2025). Modalities Contribute Unequally: Enhancing Medical Multi-modal Learning through Adaptive Modality Token Re-balancing. ICML.
> - Wang et al. (2020). What Makes Training Multi-Modal Classification Networks Hard?. arXiv.
> - Huang et al. (2021). What Makes Multi-modal Learning Better than Single (Provably). arXiv.

---

> > ### Comment · Reviewer_LV5f · 2025-08-04
> > **Question about information leakage**
> >
> > Thanks for your responses.
> >
> > Regarding the patch sizes, if you have longer patches in LFP that in spikes and split the data into random splits at levels of context windows, then there could be some information leakage. Can you please comment?

---

> > > ### Author Response · Authors · 2025-08-05
> > >
> > > We thank the reviewer for their follow-up and appreciate their engagement.
> > >
> > > This is a great question. We would like to highlight that for all signals, we apply **spatial patching only**. Thus, both LFP and spike signals used for alignment contain the **same number of timesteps**, and these signals only differ in the number of spatial patches. Before alignment, we average the embeddings of spatial patches that are at the same timestep, and the alignment objective is computed between average embeddings of LFP and spike signals at each timestep. Since patching is **strictly** along the **spatial** (channel) dimension without any overlap, **no temporal information is repeated across patches**, and thus our patching strategy **does not introduce information leakage**. We thank the reviewer for raising this important question, and we will update the text in Section 3.2 to clarify it.
> > >
> > > Regarding data splits, we first created **nonoverlapping** segments of length 3 to 5 seconds, and assigned these segments randomly to train, validation or test splits without replacement, as it is common practice and also done in NDT-2 (Ye et al., 2023; as denoted on the footnote on page 6 of their NeurIPS ‘23 manuscript, in which they used 1 second segments), which is also consistent with Neural Latent Benchmark (Pei et al., 2021) as stated there. **Thus, no timepoint is shared across train, validation, and test splits**, similar to prior works. Also, the window durations we use are substantially longer (3 to 5 times longer than the segments considered in NDT-2). Thus, our longer windows provide a more conservative choice and further reduce the risk of temporal leakage due to correlation between adjacent non-overlapping segments.
> > >
> > > We again thank the reviewer for their follow-up, and we are happy to answer any other questions they might have.
> > >
> > > References:
> > >
> > > - Pei et al. (2021). Neural Latents Benchmark ‘21: Evaluating latent variable models of neural population activity. Proceedings of the Neural Information Processing Systems Track on Datasets and Benchmarks. https://datasets-benchmarks-proceedings.neurips.cc/paper/2021/hash/979d472a84804b9f647bc185a877a8b5-Abstract-round2.html
> > > - Ye et al. (2023). Neural Data Transformer 2: Multi-context Pretraining for Neural Spiking Activity. Thirty-seventh Conference on Neural Information Processing Systems.

---

> > > ### Author Response · Authors · 2025-08-08
> > >
> > > As the discussion period draws to a close, we would like to thank the reviewer again for their feedback. We remain happy to address any further questions or concerns they might have.
> > >
> > > Additionally, we would like to bring to the reviewer’s attention our response to Reviewer KUPu, which provides further evidence that our distillation approach remains effective even when controlling for dataset size differences between spike and LFP models.

---

### Note · Authors · 2025-08-12

Dear Reviewers and Area Chair,

We sincerely thank you for your time and effort in evaluating our work. We believe the rebuttal period was highly productive, thanks to the reviewers’ constructive feedback, which has significantly strengthened our manuscript. We would like to briefly summarize the new analyses conducted during the rebuttal and discussion period:

- **Showing a stronger form of generalization** for the distillation approach by demonstrating that distilled LFP models can generalize to sessions containing only LFP signals, without any further fine-tuning, training, or distillation.
- **Analyses with a teacher MS-Spike model trained on the same sessions as MS-LFP**, validating that a substantial portion of the performance gap between spike and LFP models arises from signal differences rather than dataset size differences.
- **A new distillation with the new teacher model trained on a subset of sessions (from the previous step)**, demonstrating that the approach remains effective even when the teacher model is trained on a reduced dataset, and that *the gap between MS-Spike and MS-LFP is overcome by distillation*.
- **Comparisons to a new multimodal baseline**, namely MSID, and showing that our distillation approach also outperforms this additional baseline.
- **An ablation study on the choice of encoder architecture**, validating our initial choice of using rotary transformers.
- **A scaling experiment on the teacher MS-Spike model size**, indicating the strength of the distillation approach even when a smaller teacher model is used.
- **An ablation study on scheduled unfreezing of the teacher MS-Spike model** during the distillation procedure, confirming the advantage of our initial choice to freeze the MS-Spike teacher model throughout the distillation process.

We thank the reviewers for all their helpful comments, which we believe are addressed by the extensive analyses outlined above, together with our detailed responses. We hope that the above additional efforts will help the reviewers and area chair with their final evaluations. We once again thank the reviewers and area chair for their time, constructive feedback, and thoughtful engagement with our work.

Authors

---

### Decision · Program_Chairs · 2025-09-17

**Decision:**

Accept (poster)

**Comment:**

This paper introduces an unsupervised distillation framework that transfers representations from spike-based model to a local field potential (LFP) only model, and shows that the resulting LFP models align well with spike representations and outperform LFP-only baselines in predicting behavior. The reviewers were unanimous in their assessment that the paper makes a worthwhile contribution to the literature and should be accepted.  Congratulations! Please be sure to address all reviewer comments and criticisms in the final manuscript.